**Article** https://doi.org/10.1038/s41467-026-69760-y

# High-quality metagenome assembly from nanopore reads with nanoMDBG

Gaëtan Benoit[1], Robert James[2], Sébastien Raguideau [3], Georgina Alabone[2,3,4], Tim Goodall [5], Rayan Chikhi [1,6] & Christopher Quince [2,3,4,6] ✉

Third-generation long-read sequencing technologies, significantly improve metagenome assemblies. Highly accurate PacBio HiFi reads can yield hundreds of near-complete metagenome-assembled genomes (MAGs) from a single sample. Recently, the accuracy of the more cost-effective Oxford Nanopore Technologies (ONT) platform has increased to a per-base error rate of 1-2%. However, current metagenome assemblers are optimized for HiFi and do not scale to the large data sets that ONT enables. We present nanoMDBG, an evolution of metaMDBG, which supports the latest ONT reads through an error correction pre-processing step in minimizer-space. Across a range of ONT datasets, including a large 400 Gbp soil sample, nanoMDBG reconstructs up to twice as many high-quality MAGs as the next best ONT assembler, metaFlye, while requiring a third of the CPU time and memory. Critically, the latest ONT technology can now produce comparable MAG construction results as those obtained using PacBio HiFi at the same sequencing depth.

Metagenomics, the sequencing of genetic material recovered directly from environmental samples, enables the exploration of complex microbial communities[1,2]. The initial step in many metagenomic analyses is the assembly of sequencing reads into longer contiguous fragments or contigs. Metagenome assembly from short-read sequencing poses significant challenges, often yielding highly fragmented assemblies, particularly in diverse communities. Although binning methods can cluster short-read contigs into metagenome-assembled genomes (MAGs)[3–5], MAGs constructed from short reads frequently suffer from incompleteness, contamination and high fragmentation[6].

Long-read sequencing technologies, Oxford Nanopore Technologies (ONT) and PacBio HiFi, have significantly enhanced the quality of metagenomic assemblies. PacBio HiFi reads, characterized by their high accuracy, have made it possible to resolve hundreds of circular genomes directly from complex metagenomic samples, offering unprecedented insights into microbial communities[7–9]. While applications of earlier ONT reads to metagenomics were limited by high error rates[10], recent advances in ONT chemistry and base-calling

algorithms now enable reads with error rates around 1%[11,12], opening new avenues for method development.

MetaFlye was the first pioneering assembler specifically developed for long-read metagenomic assembly[13]. This was designed to tackle the assembly of both noisy ONT data and HiFi. Hifiasm-meta[8], in contrast, specifically harnesses the high accuracy of HiFi reads to reconstruct complete, circularized genomes and outperforms meta-Flye in this regard. More recently, we introduced metaMDBG[9], also optimized for assembling HiFi metagenome data, which reduces memory usage by an order of magnitude compared to hifiasm-meta and surpasses all current tools in both the production of near-complete MAGs and runtime efficiency. However, hifiasm-meta and metaMDBG both underperform on ONT data, even with the latest base-calling accuracy improvements. As a result, metaFlye remains the only viable option for assembling ONT datasets. This gap motivated us to develop an evolution of metaMDBG specifically designed to address the error-handling challenges in ONT data.

At the core of our algorithms is a variant of the de-Brujin graph (DBG) tailored for long reads, the minimizer-space de-Brujin graph

[1]Institut Pasteur, Université Paris Cité, Sequence Bioinformatics Unit, Paris, France. [2]Quadram Institute, Norwich, UK. [3]Earlham Institute, Norwich, UK. [4]School of Biological Sciences, University of East Anglia, Norwich, UK. [5]UK Centre for Ecology & Hydrology, Wallingford, UK. [6]These authors jointly supervised this work: Rayan Chikhi, Christopher Quince. ✉e-mail: christopher.quince@earlham.ac.uk

(MDBG), first introduced in rust-mdbg, then refined in metaMDBG[9,14]. This approach considers only a small subset (around 0.5%) of k-mers from each read, selected uniformly using a minimizer-based sampling technique. The chosen k-mers are then linked into chains of size $k\prime$ and further extended using a de Bruijn graph-like framework. In order to exploit the full potential of high-accuracy long reads, metaMDBG creates long and specific chains ($k\prime \approx 100$). However, in the presence of sequencing errors, such long chains inevitably end up containing erroneous k-mers, which break contiguity. It is also important to note in the context of metagenomics, where multiple strains of a species may potentially be present[15], that the minimizer-space assembly will not produce the finest scale of resolution possible. Due to the low density of minimizers used, strains that are sufficiently similar may be collapsed into consensus genomes.

A first attempt to deal with sequencing errors in minimizer-space was introduced in rust-mdbg[14], through read correction inspired by multiple sequence alignment. For each read represented by its minimizers, similar reads with a high fraction of shared minimizers are recruited, then aligned to generate a consensus sequence in minimizer-space using partial order alignment (POA). This method has notable limitations. Firstly, the read recruitment implemented using a resource-intensive disk-based read bucketing step degrades performance. Secondly, a low similarity threshold for read recruitment is overly permissive, especially in the metagenomics context, leading to reads from different species being potentially aligned together which in turn yields erroneously corrected reads. While a higher similarity threshold could mitigate this risk, it would also exclude genuine alignments with long flanking regions. Also, the low density of minimizers typically utilized by rust-mdbg and metaMDBG precludes accurate estimation of divergence between reads and leads to subpar recruitment.

We introduce nanoMDBG, a novel algorithm integrated into the metaMDBG software, which revisits the concept of read correction in minimizer-space to address errors in ONT data. At the heart of nanoMDBG is an innovative use of fast seed-and-chaining techniques for read correction. It enables the rapid alignment and pileup of minimizers from similar reads to efficiently build consensuses. We further optimize the resource-intensive all-vs-all read mapping phase, a common bottleneck in read correction, by leveraging variable minimizer densities. Specifically, a low-density setting allows for quick recruitment of the most similar reads needed for effective correction, while a high-density setting ensures precise estimation of sequence divergence among the recruited reads.

This improved error correction allows nanoMDBG to handle large ONT datasets efficiently and accurately. For instance, on the largest tested dataset, a 400 Gbp soil sample, the correction only took 16 h using 32 cores (10% of overall assembly time) and 74 GB of memory (a fraction of the subsequent assembly requirements). On this dataset, nanoMDBG substantially outperformed state-of-the-art assemblers, recovering 201 more near-complete MAGs than metaMDBG and 144 more than metaFlye. When using PacBio HiFi data from the same samples, we demonstrate that nanoMDBG can now deliver ONT-derived results comparable in MAG number to those achieved with HiFi sequencing.

## Overview of minimizer-space assembly with nanoMDBG

We present nanoMDBG, a metagenome assembly method within metaMDBG, designed for Oxford Nanopore Technologies (ONT) long-read data. Given a set of input reads, nanoMDBG produces a FASTA file containing assembled contigs. An overview of the assembly workflow is illustrated in Fig. 1. The universal minimizers, which are k-mers that map to an integer below a fixed threshold (see "Methods"), are first identified in each read. Each read is thus represented as an ordered list of the selected minimizers, denoted as an mRead. To improve accuracy, mReads are corrected using a multiple sequence alignment approach, employing a seed-and-extend strategy to align minimizers. For a given target mRead to correct, up to twenty of the most similar mReads are recruited, using a low density of minimizers to speed up the alignment process (20% of the minimizers in each mRead by default). The recruited mReads are then re-aligned to the target mRead using the complete minimizer set, refining divergence estimation and reducing recruitment of reads from unrelated genomes. These alignments are then used to pileup minimizers using a minimizer-space variation graph, and a consensus sequence is extracted from the most supported path. The corrected mReads are then assembled into mContigs using the metaMDBG assembler. These mContigs are then converted back into base-space and polished.

## Results

We first evaluate assembly results on ONT data only. We then compare assemblies obtained from ONT and HiFi data generated from the same samples and conclude with an evaluation of assembly errors across both platforms.

### Evaluation of nanoMDBG for ONT metagenome assembly

We compared nanoMDBG (v1.2) to two state-of-the-art assemblers: metaFlye (v2.9.3-b1797) and metaMDBG (v1.2) on one mock community and three real metagenomes (see Table 1). The commands that were used are provided in Supplementary Table 1, and the assembly results are summarized in Supplementary Tables 2, 3, and 4. A comparison to hifiasm-meta (v0.3-r063.2) is also given, although only on a subset of the data sets as explained below. The mock community, "Zymo", contains 21 species for which reference genomes and abundances are known. The first real metagenome, "Human gut", is a 50 Gbp Human gut sample. The second metagenome, "Zymo Fecal Reference", is a 200 Gbp reference standard constructed by pooling fecal samples from multiple donors[16]. The third data set, "Soil", is a 400 Gbp soil sample taken from an agricultural field in Oxfordshire (UK)—see "Methods". All data sets were sequenced with the latest R10.4.1 ONT flow cells, and reads were subsampled after filtering by selecting the first N reads necessary to achieve the specified data set sizes. This was done to allow comparison with equivalent-sized HiFi PacBio data sets below. We note that the Human gut and Soil datasets were generated for this study, while the Zymo Fecal Reference dataset is publicly available.

We first evaluated the assemblers on the Zymo mock community using MetaQUAST (see "Methods") to compare assemblies against the available reference genomes (see Supplementary Table 2) and determine their accuracy. The mock community contains 21 strains, including 5 E. coli strains of similar abundance. Four rare species (coverage ≤6x) could not be fully reconstructed by any assembler, and two yeast genomes were absent from all assemblies. Among the abundant, non-E. coli species, nanoMDBG and metaFlye performed similarly in both the number of species recovered as single contigs and in consensus quality. Each recovered 4 genomes as circular contigs and 2 as single linear contigs. Both produced ≈ 6 mismatches and ≈ 4 indels per 100 kbp, as well as a total of 9 misassemblies. MetaMDBG generally yielded slightly more fragmented assemblies with more errors. Hifiasm-meta produced a highly fragmented assembly with a large number of misassemblies, likely because it is optimized for high-accuracy PacBio HiFi data; therefore, we excluded it from further comparisons in this section. No assembler could fully resolve the E. coli strain diversity. The highest completeness for an E. coli strain was achieved by hifiasm-meta (95%) but with 34 misassemblies, followed by MetaMDBG (88.5% with 21 misassemblies), nanoMDBG (82.4% with 17 misassemblies), and metaFlye (46.4% with 2 misassemblies).

In the case of the real microbiome data sets we used SemiBin2[4] (v.2.1.0) to bin contigs obtained by each of the assemblers and evaluated the quality of the resulting bins with CheckM2[17] (v.1.0.1). We

## a  NanoMDBG strategy overview

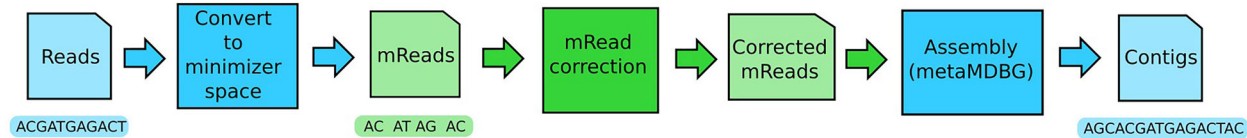

## b  Seed-and-chaining as an minimizer-space alignment method

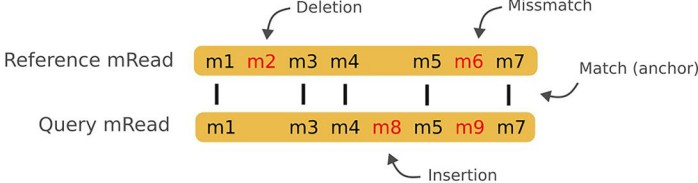

## c  Read correction in minimizer-space

### 1) Ultra-fast mRead recruitement using low-density minimizer-space alignment

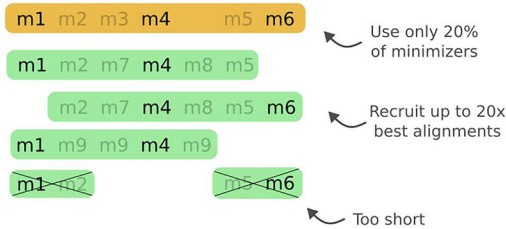

### 2) Accurate high-density alignment filtering

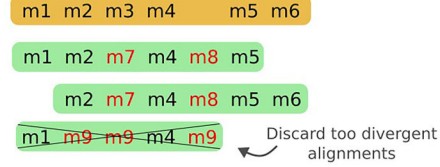

### 3) Derive minimizer-space variation graph from high-density alignments

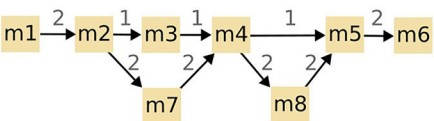

### 4) Generate consensus

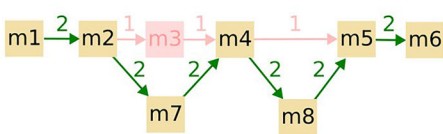

**Fig. 1 | Overview of the algorithmic steps of nanoMDBG. a** Overview of nanoMDBG workflow. Steps in green correspond to our new contributions. **b** Illustration of an alignment at the level of minimizers. **c** Overview of the read correction method in minimizer-space. **1**, For a given target read to correct (in yellow), the most similar reads are recruited (up to 20x coverage) using seed-and-chaining alignment. We use only 20% of the read minimizers (by lowering the threshold for selecting universal minimizers) to speed up this costly recruitment process. **2**, Using the full read minimizers, we re-align the recruited reads using seed-and-chaining in order to accurately estimate their divergence with the target read. Reads with divergence over 4% are discarded. **3**, Using the accurate alignment computed in the previous step, we pile up the minimizers using a simple variation graph. The graph is initiated with the minimizer from the target read (path on top of the graph). Recruited reads are then added sequentially. A mismatch in a recruited read creates a path parallel to a node from the target read (node m7), while an insertion creates a path between two nodes from the target read (node m8) **4**. The most supported path (in green) is extracted from the graph and used as a consensus for the target read.

grouped bins into conventional categories based on their level of completeness and contamination reported by CheckM2: near-complete MAGs ( > 90% completeness and < 5% contamination), high-quality MAGs ( > 70% completeness and < 5% contamination) and medium-quality ( > 50% completeness and < 5% contamination). We

also report the number of near-complete MAGs in a single contig (scMAGs) after binning.

For all three tested real communities, nanoMDBG reconstructed appreciably more near-complete MAGs than the other assemblers (see Fig. 2a and Supplementary Tables 3 and 4). From the Human gut

**Table 1 | Evaluated metagenome datasets**

| Sample | Accessions | # bases (Gb) | N50 read length (kb) | Average quality score | Sample description |
|---|---|---|---|---|---|
| **Oxford Nanopore datasets** | | | | | |
| Zymo | ERR15316007 | 12 | 5.9 | 21.2 | Mock community ZymoBIOMICS D6331 |
| Human gut | ERR15285694 | 50 | 7.8 | 19.6 | Human gut sample |
| Zymo Fecal Reference | - | 206 | 12.2 | 17.4 | Reference standard constructed by pooling fecal samples from multiple donors |
| Soil | ERR15289757 | 400 | 9.8 | 20.5 | Agricultural field in Oxfordshire (UK) |
| **HiFi PacBio datasets** | | | | | |
| Human gut | ERR15289675 | 50 | 8.9 | 27 | Human gut sample |
| Zymo Fecal Reference | - | 255 | 8 | 28.8 | Reference standard constructed by pooling fecal samples from multiple donors[16] |
| Soil | ERR15289804 | 250 | 10 | 26.4 | Agricultural field in Oxfordshire (UK) |

Two samples were previously publicly available (ONT Zymo Fecal Reference and HiFi Zymo Fecal Reference, see "Data availability" section), and we newly sequenced five samples (ONT Zymo, ONT Human gut, ONT Soil, HiFi Human gut and HiFi Soil). To facilitate comparison across sequencing technologies, these datasets were subsampled by selecting the first N reads until the desired data volume was reached. Statistics were computed with the command "seqkit stats --all"[50].

sample, nanoMDBG reconstructed 78 near-complete MAGs (23 more than metaMDBG and 19 more than metaFlye), 255 from the Zymo Fecal Reference dataset (30 more than metaMDBG and 88 more than metaFlye) and 260 from the Soil sample (201 more than metaMDBG and 144 more than metaFlye). The number of scMAGs generated by nanoMDBG is substantially improved compared to the other assemblers. NanoMDBG reconstructed 26 scMAGs from the Human gut sample (15 more than metaFlye), 83 from the Zymo Fecal Reference dataset (39 more than metaFlye) and 91 from the Soil sample (79 more than metaFlye). As a further validation of the quality of the near-complete MAGs, we predicted the presence of rRNA and tRNA genes (see Methods). The near-complete MAGs generated by each assembler usually do contain the expected complement of RNA genes (91% for nanoMDBG and metaFlye, and 87% for metaMDBG). We also note that the majority of the near-complete MAGs produced by all assemblers contained fewer than 10 contigs (see Supplementary Fig. 1).

The improvement in assembly quality by nanoMDBG resulted in MAG collections that mapped a larger fraction of reads from each dataset (see Fig. 2b) and hence, are more representative of the community. For the Human gut dataset, nanoMDBG was the only assembler to map more than 50% of reads to near-complete MAGs (ncMAGs). As the complexity of the datasets increased, the fraction of mapped reads decreased. On the Zymo Fecal Reference, 38% of reads were mapped to near-complete nanoMDBG MAGs, but only 7% on the Soil near-complete MAGs. The fraction of reads mapping to ncMAGs cannot be larger than the fraction of reads mapping to the overall assembly. If genomes have very low coverage, then they will not be assembled at all. It is useful, therefore, to compare these percentages to the fraction of reads that map onto the entire nanoMDBG assemblies, which gives their upper limits: Human gut: 82%, Zymo Fecal Reference: 78% and Soil: 48%. Even accounting for the fact that not all genomes will be from prokaryotes, this suggests that we are underperforming in the Soil data set.

To further investigate whether the low fraction of mapped reads in the Soil dataset was due to insufficient sequencing depth or assembly failure, we compared the number of recovered MAGs to the actual species diversity in the assembly. This was estimated by identifying single-copy core genes (SCGs) from contigs and clustering them at 97% identity (see "Methods") (see Fig. 3). Based on SCG analysis, we estimated that the Human gut dataset contained 129 species, the Zymo Fecal Reference dataset 481 species, and the Soil dataset 3772 species. In the Human gut dataset, 70% of predicted species had high coverage ( > 10×), and 85% of those were converted into MAGs. In the Zymo Fecal Reference dataset, over 90% of abundant species ( > 10× coverage) were also converted into MAGs. However, in the Soil

dataset, only 40% of predicted species had coverage > 10x, and just 37% of those abundant species were converted into MAGs. The low fraction of mapped reads in the Soil dataset can therefore be attributed to both the large species richness and the limited sequencing depth relative to that diversity, as well as challenges in assembling abundant genomes.

To summarize the microbial diversity from the Soil sample, we constructed a phylogenetic tree with leaves as genera (see "Methods") for all near-complete MAGs from all assemblers (see Fig. 2c). We note that amongst the total 435 retrieved near-complete MAGs, only 5 could be assigned to a known species with a reference genome (ANI > 95%). The improved MAG recovery by nanoMDBG translates into a more representative picture of microbial diversity at all levels of evolutionary divergence compared to the other assemblers tested. In total, we observed 84 genera that were recovered from the Soil data sets by nanoMDBG but are missing from the near-complete MAG collections of the other programs. When the other assemblers did recover MAGs from the same genus, nanoMDBG usually found more MAGs. Finally, we can see large parts of the tree that are represented by only nanoMDBG MAGs; indeed, one phylum (41 families) was found only by nanoMDBG, compared to 12 families specific to metaFlye and one family specific to metaMDBG (see Fig. 2d and Supplementary Table 7).

The assembly of phage genomes and plasmids potentially present in metagenomes can be particularly challenging. To compare phage and plasmid reconstruction between assemblers, we used geNomad[18] to identify circular contigs that were potential plasmid and phage genomes (see Supplementary Table 5). On the Human gut and Zymo Fecal Reference samples, metaFlye reconstructed more viruses than nanoMDBG (3 and 60 more, respectively), but nanoMDBG found 237 more phages in the Soil data set. We used CheckV[19] to assess the completeness of circular phages. We found that 61% of genomes recovered by nanoMDBG were judged high-quality against 57% for metaFlye. NanoMDBG found substantially more circular plasmids compared to metaFlye for all three metagenomes (9 more on the Human gut sample, 24 more on the Zymo Fecal Reference and 9 more on the Soil sample). MetaMDBG usually obtained fewer circular viruses and plasmids than nanoMDBG and metaFlye.

Computational scalability, given the potentially large size of ONT metagenome data sets, is an important consideration. By performing the correction step in minimizer-space, our novel method, nanoMDBG, naturally extends metaMDBG to ONT data without impacting overall performance. For example, on the 400 Gbp Soil sample, the correction step required only 16 h using 32 cores (accounting for 10% of the total assembly time) and 74 GB of memory, which is a fraction of the memory required for subsequent assembly (see Supplementary

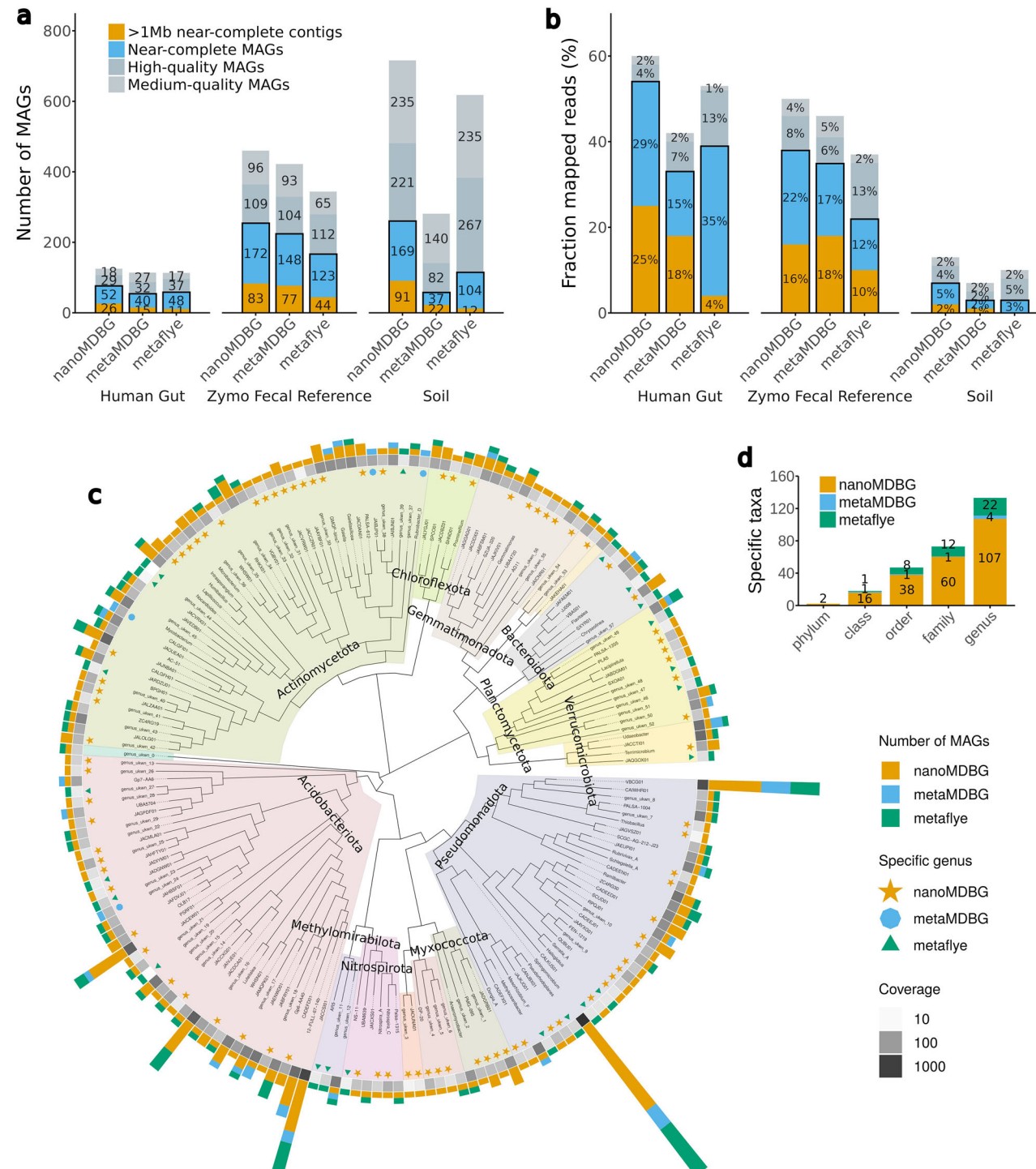

**Fig. 2 | Assembly results on the three ONT metagenomics data sets. a** CheckM2 evaluation. A MAG is "near-complete" if its completeness is ≥90% and its contamination is ≤5%, "high-quality" if completeness ≥70% and contamination ≤5%, "medium quality" if completeness ≥50% and contamination ≤5%. **b** The percentage of mapped ONT reads on MAGs. **c** Phylogenetic tree of genera recovered from the ONT soil data set for all assemblers combined. For the near-complete bacterial MAGs, we generated a de novo phylogenetic tree based on GTDB-Tk marker genes, displayed at the genus level. The outer bar charts give the number of MAGs found in each genus. The colored symbols then denote genera recovered by only one of the assemblers. The grayscale heat map illustrates the aggregate abundance of dereplicated MAGs in a genus. **d** Number of taxa at different levels that are unique to each assembler.

Table 6). NanoMDBG completed the full Soil dataset assembly in just 6 days, compared to 28 days for metaFlye. In terms of memory efficiency, nanoMDBG used 125 GB, while metaFlye required 753 GB, making the latter potentially inaccessible for many laboratories. Both assemblers efficiently handled the 50 Gbp Human gut sample, with

nanoMDBG completing the task in 5 h and metaFlye in 8 h. However, nanoMDBG used only 17 GB of memory compared to metaFlye's 115 GB, enabling nanoMDBG to potentially assemble datasets of equivalent complexity on a laptop, which could be important given the portability of ONT sequencing devices.

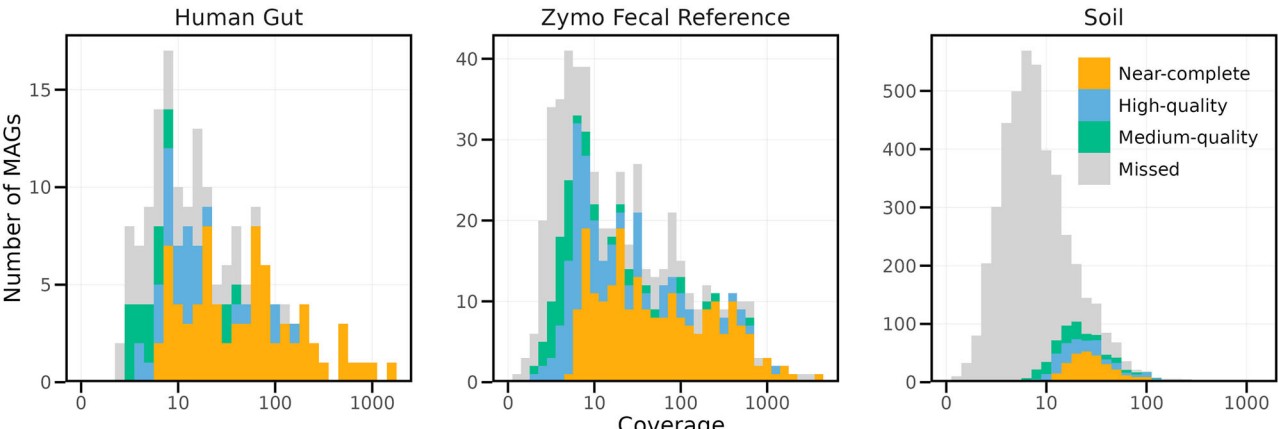

**Fig. 3 | Rank abundance plot of species identified across the three ONT samples.** Species counts were estimated based on the presence of the single-copy core gene COG0060 in nanoMDBG contigs, based on 97% identity clustering (see "Methods"). Colors indicate MAG quality, with "missed" representing species where the SCG was detected but not assigned to a MAG.

## Evaluation of ONT base-level read correction tools before assembly

We explored an alternative approach for processing ONT data by applying read correction tools prior to assembly. Specifically, we assessed two recent haplotype-aware correction tools: HERRO[20] and DeChat[21]. While DeChat is tailored for metagenomics, we note that HERRO was developed for genomic data, with the authors recommending its use on reads longer than 10 kbp, higher than the typical N50 read length of metagenomic samples.

We applied HERRO (through the Dorado ONT base calling pipeline) and DeChat to the 50 Gbp Human gut sample, assembled the corrected reads using each assembler, and evaluated the quality of the recovered MAGs (see Supplementary Fig. 2). Assemblies incorporating HERRO consistently yielded fewer MAGs compared to raw assemblies. This likely stems from HERRO's inability to effectively process reads shorter than 4 kbp, which are prevalent in metagenomic datasets. In contrast, DeChat substantially enhanced assembly outcomes: it improved the recovery of scMAGs by metaMDBG (11 additional scMAGs and 9 near-complete MAGs) and metaFlye (1 additional scMAG and 8 near-complete MAGs). For nanoMDBG, DeChat resulted in 1 additional scMAG, although raw assemblies with nanoMDBG alone still produced more near-complete MAGs overall (2 additional near-complete MAGs).

Although read correction can enhance assembly quality, it comes with substantial computational costs. For instance, HERRO required 56 h (GPU mode) and 210 GB of memory, while DeChat took 30 h and 250 GB. In contrast, nanoMDBG completed the entire assembly process, including contig polishing, in just 5 h with 17 GB of memory. These computational costs prevented us from evaluating the read correction tools on the larger and more complex Zymo Fecal Reference and Soil data sets.

## Evaluation of metagenome assembly quality from ONT and PacBio HiFi data

To enable a direct comparison between ONT and HiFi assembly quality, we generated HiFi data from the same samples used for ONT sequencing. Specifically, we produced 50 Gbp of PacBio HiFi data from the Human gut sample and 250 Gbp from the Soil sample, and we utilized a publicly available 255 Gbp HiFi dataset for the Zymo Fecal Reference sample[16] (see Table 1). We ran three HiFi metagenome assemblers: metaMDBG (v1.2), metaFlye (v2.9.3-b1797), and hifiasm-meta (v0.3-r063.2). We did not apply nanoMDBG error correction to the HiFi data because HiFi reads, after homopolymer compression, a standard preprocessing step in HiFi assembly, are nearly error-free.

Figure 4a presents the number of near-complete MAGs and scMAGs recovered using both sequencing technologies at varying sequencing depths (selecting the first N reads until the desired data volume was reached). Note that different average read lengths may result in differing read numbers at the same depth when comparing between technologies, but this is, in our opinion, the most valid comparison. For clarity, the figure only shows results obtained with metaMDBG on HiFi data and nanoMDBG on ONT data, the combinations that consistently produced the best assemblies across experiments. Comprehensive results for all assemblers are provided in Supplementary Fig. 3.

On the Human gut sample, ONT data yielded more near-complete MAGs at lower sequencing depths (30 additional MAGs at 10 Gbp of data). At this low depth, most HiFi MAGs were of high quality. At sequencing depths exceeding 20 Gbp, HiFi consistently recovered more near-complete MAGs compared to ONT data, with 36 more near-complete MAGs at 50 Gbp (with 12 more as single-contigs). For the Zymo Fecal Reference dataset, more near-complete MAGs were reconstructed from HiFi technologies at all depths, with 294 MAGs recovered from HiFi compared to 255 from ONT at 200 Gbp. ONT data produced more scMAGs at lower depth (14 more scMAGs at 100 Gbp), but more were produced from HiFi data at 200 Gbp (5 more scMAGs). In the Soil sample dataset, the number of near-complete MAGs was comparable between the technologies, with 165 near-complete MAGs from HiFi and 166 from ONT at 250 Gbp. Nonetheless, HiFi consistently produced more scMAGs, with one additional scMAG at 50 Gbp and 32 more at 250 Gbp.

To evaluate differences between near-complete MAGs generated by the two technologies, we used the dRep[22] tool to identify shared and specific MAGs and assessed their coverage and contiguity (see Fig. 4b). A total of 333 near-complete MAGs were shared between the technologies, while 221 and 126 MAGs were unique to HiFi and ONT, respectively. Shared MAGs generally exhibited higher coverage (median ≈ 35× vs. ≈ 17× for MAGs specific to HiFi and ≈ 25× for MAGs specific to ONT) and were reconstructed in fewer contigs compared to those unique to a single technology. Unshared MAGs, therefore, represent the most challenging to assemble.

Supplementary Fig. 3 explores the recovery of circular viruses and plasmids across assemblers (see also Supplementary Table 5). For the Human gut sample, the number of phages and plasmids recovered was comparable between technologies (at 50 Gbp; 32 phages with HiFi using hifiasm-meta and 27 with ONT using metaFlye; 42 plasmids with HiFi using metaMDBG and 36 with ONT using nanoMDBG). In the Zymo Fecal Reference dataset, ONT consistently recovered more

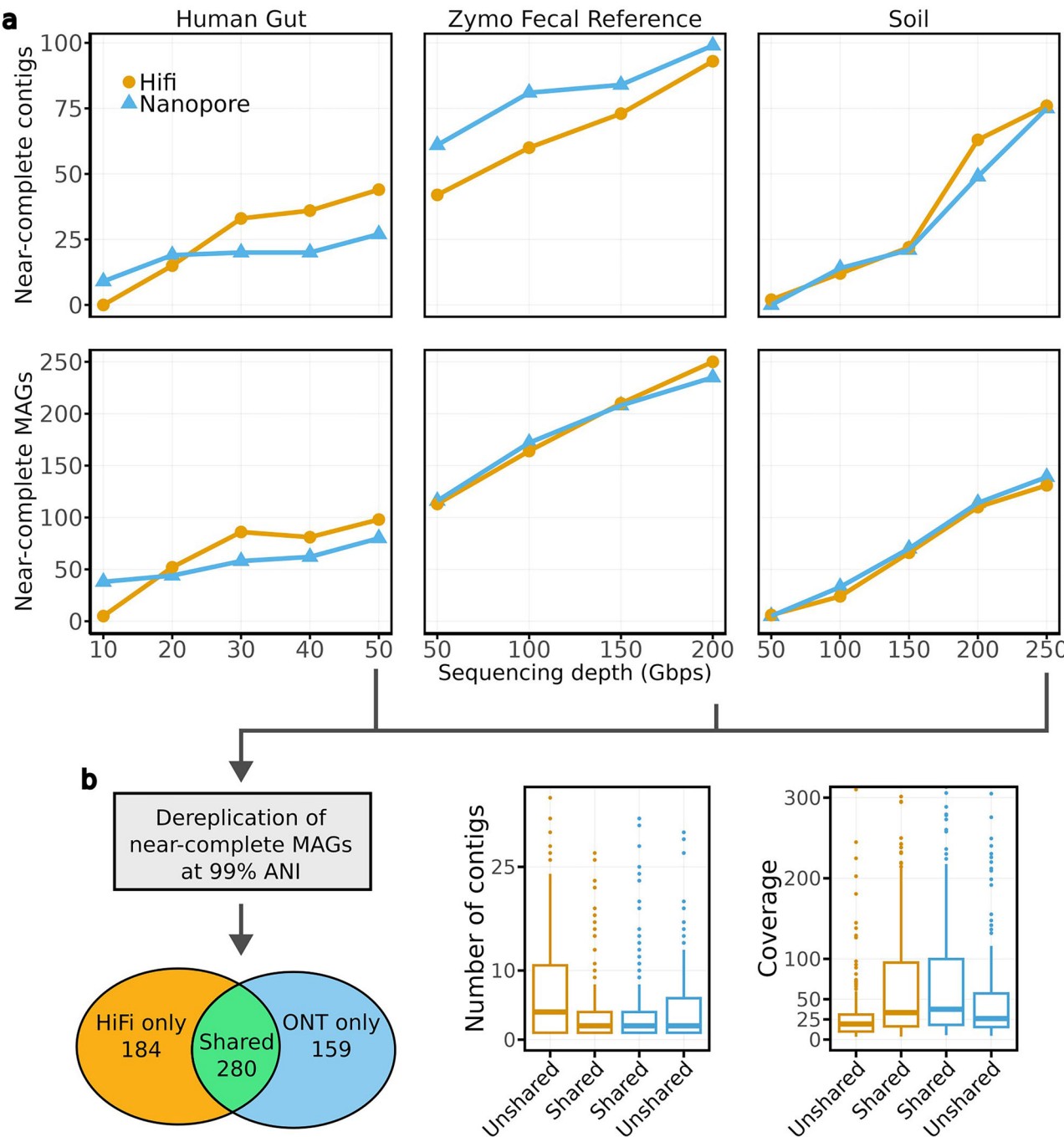

**Fig. 4 | Assembly results from ONT and HiFi data generated from the same samples.** For each data type, we show the results of the assembler that obtained the best results consistently: metaMDBG for PacBio HiFi data and nanoMDBG for ONT data. **a** CheckM2 evaluation with respect to sequencing depth. We note that the number of near-complete contigs (top plots) is included in the total number of near-complete MAGs (bottom plots). **b** Statistics of near-complete MAGs recovered by both and specific to each data type. We de-replicated MAGs per dataset using the highest sequencing depth available for each dataset (50 Gb, 200 Gb and 250 Gb, respectively). The boxplot elements are the median (horizontal bar), 25th and 75th percentiles (box limits Q1 and Q3), Q1-1.5*IQR and Q3+1.5*IQR (whiskers, IQR=Q3-Q1) and outliers. Summary statistics in **b** (n, min, max, median, 25th and 75th percentiles, lower whisker, upper whisker): Number of contigs–Unshared HiFi (221, 1, 63, 5, 2, 11, 1, 22); Shared HiFi (333, 1, 66, 2, 1, 5, 1, 11); Shared ONT (333, 1, 70, 3, 1, 6, 1, 13); Unshared ONT (126, 1, 64, 6, 2, 10, 1, 22): Coverage–Unshared HiFi (221, 3.6, 567.8, 16.7, 9.8, 26.8, 3.6, 50.2); Shared HiFi (333, 3.8, 2834.3, 33.8, 16.7, 100.4, 3.8, 221.4); Shared ONT (333, 4.6, 3548.8, 34.4, 17.5, 102.7, 4.6, 229.1); Unshared ONT (126, 4.4, 1068.1, 25.6, 16, 47.5, 4.4, 91.2).

phages and plasmids at all sequencing depths (At 200 Gbp, 121 phages and 90 plasmids were recovered from ONT data compared to 93 phages and 83 plasmids from HiFi). For the Soil dataset, HiFi recovered more viruses at all depths (572 with HiFi using metaMDBG vs. 418 with ONT using nanoMDBG). The recovery of plasmids was also higher from HiFi data (20 plasmids compared to 13). Notably, metaMDBG and

nanoMDBG were the most effective assemblers for recovering circular plasmids on HiFi and ONT data, respectively. For circular phages, metaMDBG and hifiasm-meta excelled with HiFi data, while metaFlye or nanoMDBG performed better on ONT data, depending on the sample complexity.

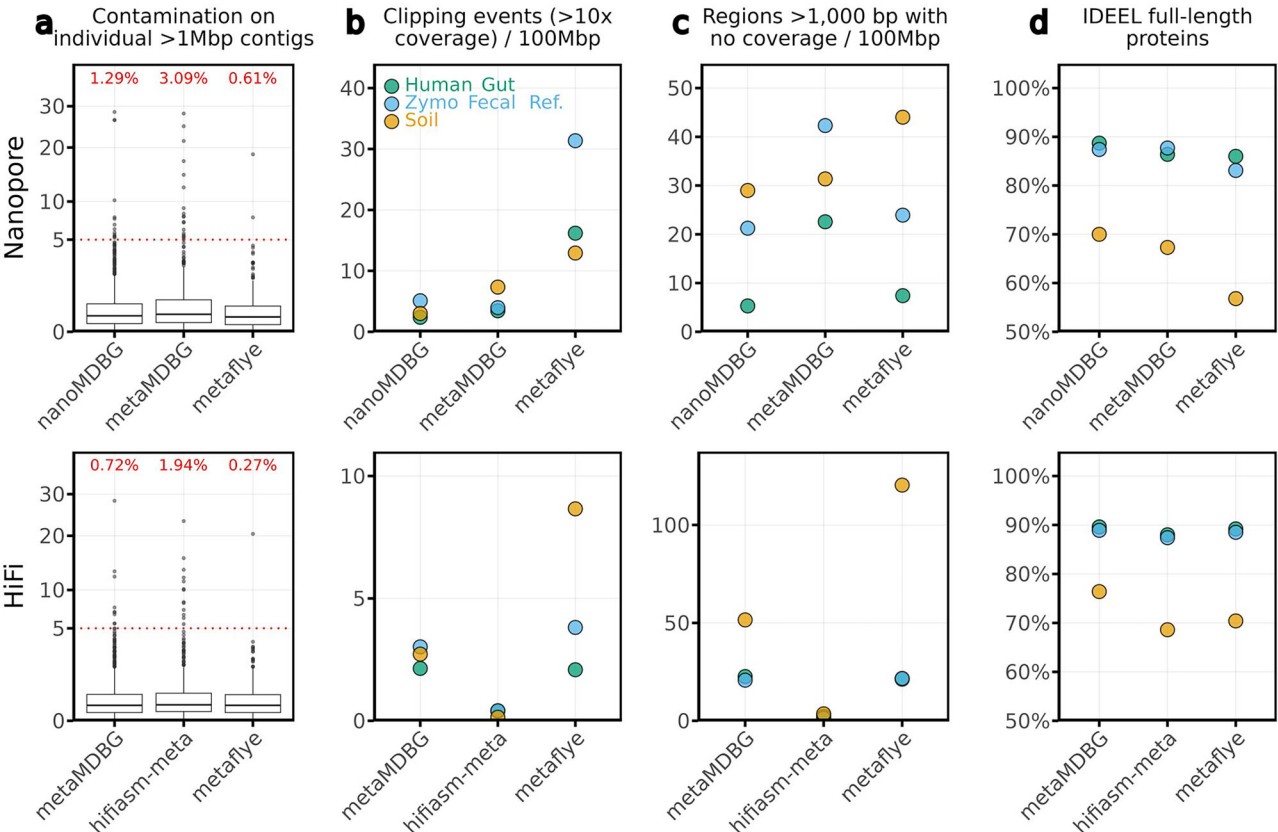

**Fig. 5 | Assembly errors across assemblers.** The top row presents results for Nanopore data, and the bottom row for HiFi data. **a** Contamination level in contigs longer than 1 Mbp. The percentage above each plot indicates the overall proportion of long contaminated contigs (with contamination over 5%). **b** Number of clipping events supported by at least 10 reads. **c** Number of regions over 1,000 bp with no apparent read coverage. **d** Proportion of predicted proteins which are ≥90% the length of their best-matching known protein in a reference database (from contigs with more than 10x coverage). In (**a**), results are aggregated across assemblies of the three data sets: Human Gut, Zymo Fecal Reference and Soil, but separated by both data set and assembler in **b**, **c**, and **d**. The boxplot elements are the median (horizontal bar), 25th and 75th percentiles (box limits Q1 and Q3), Q1-1.5*IQR and Q3+1.5*IQR (whiskers, IQR = Q3 − Q1) and outliers. Summary statistics in (**a**) (n, min, max, median, 25th and 75th percentiles, lower whisker, upper whisker): Nanopore-nanoMDBG (1006, 0, 28.47, 0.15, 0.04, 0.46, 0, 1.09); metaMDBG (549, 0, 28.07, 0.18, 0.05, 0.6, 0, 1.37); metaFlye (326, 0, 18.56, 0.13, 0.03, 0.39, 0, 0.92): HiFi-metaMDBG (1239, 0, 28.27, 0.14, 0.04, 0.41, 0, 0.96); hifiasm-meta (667, 0, 23.3, 0.15, 0.05, 0.445, 0, 1.03); metaFlye (362, 0, 20.41, 0.14, 0.04, 0.4, 0, 0.93).

## Evaluation of metagenomics assembly errors

Due to the complexity of metagenomic data, assemblies are inevitably prone to various types of errors. On real data sets where the ground truth is not known, it is standard to assess assemblies in terms of MAGs recovered, as we have done above. However, it is also important to evaluate assemblers in terms of the frequencies of likely artefacts, as has been recently suggested[23]. In this section, we compare assemblers across the ONT and HiFi PacBio data sets in terms of four such artefacts:

- Long contaminated contigs: the level of contamination for contigs longer than 1 Mbp and the proportion of contaminated contigs with CheckM2 contamination exceeding 5%. It is possible that some such contigs may represent biological novelty, but in general, contamination at this level likely indicates misassembly, generating a chimeric contig from more than one genome.
- Clipping events: locations within the assembly supported by at least 10 reads where 100% of the reads are clipped. This indicates a region where assembly has incorrectly joined distant sequences.
- Zero-coverage regions: regions longer than 1,000 bp within the assembly with no apparent read support. These are unambiguous errors.
- Frameshift errors: we used the program IDEEL[24] on contigs longer than 5 kb and with coverage over 10x to quantify the fraction of predicted proteins that were ≥90% the length of their best-matching known protein (higher fractions being better).

Insertion-deletion errors from sequencing noise can lead to predicted proteins that are shorter than expected due to frameshifts, leading to mismatched amino acids and stop codons. Some of these may be genuine pseudogenes, but again, in general, these will be errors.

We benchmarked three metagenome assemblers: nanoMDBG/metaMDBG (v1.2), metaFlye (v2.9.3-b1797), and hifiasm-meta (v0.3-r063.2), using both Nanopore and HiFi data. The datasets used included: 50 Gbp from the Human gut sample, 100 Gbp from the Zymo Fecal Reference sample, and 250 Gbp from the Soil sample. We did not use the full Zymo Fecal Reference dataset, as hifiasm-meta repeatedly crashed on the full dataset. We used the scripts "anvi-script-find-mis-assembly" from the Anvi'o platform[25] for measuring clipping events and zero-coverage regions (using minimap2[26] to map reads back to the assemblies with the same commands as described in ref. 23). The commands that were used are provided in Supplementary Table 1, and all results are summarized in Supplementary Tables 8 and 9.

Figure 5a shows that long contigs from all assemblers generally had low contamination (median contamination ≈ 0.35%). On ONT data, metaFlye produced the fewest contaminated long contigs: 2 out of 326 (0.61%), compared to 13 out of 1006 (1.29%) for nanoMDBG. On HiFi data, metaFlye again showed the lowest contamination (1 out of 362 long contigs; 0.27%), followed by metaMDBG (9 out of 1239; 0.72%) and hifiasm-meta (13 out of 667; 1.94%). We did not observe

strong trends based on sequencing technology or sample complexity; assemblers that produced more long contigs tended to also have a higher number of contaminated ones. Using metaMDBG on uncorrected ONT data usually leads to more contamination and errors overall.

For clipping events and zero-coverage regions, hifiasm-meta produced the cleanest assemblies, with fewer than 1 clipping event per 100 Mbp on all datasets and at most 3.5 zero-coverage regions per 100 Mbp in the Soil dataset (see Fig. 5b, c). NanoMDBG and metaMDBG performed better than metaFlye. On ONT data, nanoMDBG had up to 5 clipping events per 100 Mbp, whereas metaFlye exhibited 12 to 31 clipping events per 100 Mbp, with the highest count on the Zymo Fecal Reference dataset. For HiFi data, both metaMDBG and metaFlye produced fewer clipping events (up to 3 and 8.6 per 100 Mbp, respectively). In terms of zero-coverage regions on ONT data, nanoMDBG produced 5.3 regions per 100 Mbp in the Human gut sample and up to 29 in the Soil sample, compared to 7.4 and 44 for metaFlye, respectively. On HiFi data, metaMDBG and metaFlye showed similar performance for the Human gut and Zymo datasets ($\approx$ 20.5 regions per 100 Mbp), but metaFlye had substantially more such regions in the Soil sample (120 vs. 51 for metaMDBG).

IDEEL results indicated that nanoMDBG assemblies contained a higher fraction of full-length proteins than metaFlye (see Fig. 5d), particularly on the Soil dataset (88.7% vs. 86.0% in the Human gut, 87.4% vs. 83.1% in Zymo Fecal Reference, and 70.0% vs. 56.8% in Soil). On HiFi data, all assemblers performed similarly on the Human gut and Zymo Fecal Reference datasets ($\approx$ 88% full-length proteins). However, metaMDBG outperformed the others on the Soil sample (76.4% vs. 70.4% for hifiasm-meta and 68.6% for metaFlye). HiFi assemblies generally yielded more full-length proteins compared to ONT data, particularly on the Soil sample (70.0% from ONT using nanoMDBG vs. 76.4% from HiFi using metaMDBG).

## Discussion

We have introduced nanoMDBG, a new metagenome assembly method within metaMDBG, designed for ONT long reads. Our aim was to improve upon the metaMDBG assembler by addressing the relatively higher error rates of ONT data while simultaneously preserving the high scalability, completeness and accuracy that metaMDBG achieved. We succeeded in this goal, as the read correction pre-processing step that we added in nanoMDBG only takes 10% of the total assembly run time on the large 400 Gbp Soil sample. Our new correction module greatly improved the number of reconstructed MAGs, phages and plasmids compared to metaMDBG. Across all tested datasets, nanoMDBG outperformed metaFlye in the reconstruction of near-complete MAGs, particularly single-contig ones, with the gap in assembly quality especially pronounced in complex communities. Importantly, we demonstrate that thanks to this advance, the latest ONT sequencing technology now produces results almost comparable to PacBio HiFi sequencing at equivalent sequencing depth, despite the remaining differences in raw read accuracy. On the gut data sets at higher sequencing depth, some advantage for HiFi remains in terms of overall MAGs recovered, approximately 15% more overall, but for soil only in terms of the fraction of near-complete MAGs that are single-contig.

Given the relative cost-effectiveness of ONT reads our effective and computationally efficient assembler could prove transformative to metagenomics research. It enables the possibility of large-scale multi-sample long-read metagenomics for simpler communities such as the human gut, which will be critical for applications such as strain tracking[15] and a fine-scale understanding of evolutionary differences over time and space. For complex communities such as soil, it could be even more important, enabling for the first time large numbers of complete genomes to be generated from single soil samples cost-effectively.

However, we have also demonstrated that these results could potentially be improved. We obtained a much smaller fraction of high coverage genomes ( > 10x) that should be accessible actually as MAGs for soil, 37% vs. 85% for the Human gut. We are not certain of the reasons for this; it may be strain variation, but it does suggest the possibility of improved assembly strategies for these hyper-diverse communities.

We also evaluated the frequency of assembly artefacts produced by the different assemblers for both ONT and HiFi PacBio sequences [23]. There were a few clear patterns here, but the string graph assembler, hifiasm-meta, operating in sequence space produced far fewer clipping events and zero-coverage regions on the HiFi data than metaMDBG or metaFlye. Whereas in terms of contaminated and hence potentially chimeric contigs, hifiasm-meta had a somewhat higher rate than metaflye or metaMDBG. Otherwise on ONT reads metaMDBG and metaFlye had similar overall error rates across the different error types. Notably, with these high-accuracy R10.4 ONT reads, the truncation of predicted proteins by frameshift errors was no more prevalent than for HiFi, at least for the relatively high coverage human gut data sets. We did see more frameshift errors for ONT than HiFi in the Soil data sets, and here nanoMDBG had a slight advantage over metaFlye. Perhaps reflecting improved error correction, but in general, frameshift errors were much reduced compared to earlier iterations of ONT[24]. That errors exist in long-read metagenome assemblies is unsurprising in our opinion, given the complexity of the metagenomics assembly task. In the future, though, with algorithmic advances, it may be possible to reduce error rates without sacrificing performance or overall MAG recovery. It would certainly be desirable.

In general, the philosophy behind nanoMDBG is to try to obtain as many species or subspecies consensus genomes as possible. This is perhaps different from metaFlye, which may be attempting to reconstruct each strain separately. Our belief is that, given a comprehensive collection of consensus genomes then the use of dedicated strain-resolution tools that resolve fine-scale variation following mapping onto those genomes [27,28] or assembly graphs[29], is likely to prove the most effective way to fully resolve strain-diverse communities. However, this remains to be demonstrated and fully strain-aware metagenomics assembly may be an alternative strategy[30].

In addition to allowing the analysis of large-scale data sets, a highly efficient ONT assembler has other advantages. It makes assembly of smaller data sets feasible on for example a laptop. This aligns well with the portability of ONT sequencing, opening possibilities for on-site assembly directly in remote or field settings. In future, it may prove useful to implement a version of our algorithm that can perform streaming assembly, assembling reads in real-time as they are sequenced, the high efficiency of the underlying minimizer based denoising and assembly approach could make this feasible.

In this paper, we introduce an application, nanoMDBG, that combines read error correction and assembly in the minimizer space. Our approach is far more efficient than correcting reads at the base level, i.e., sequence read error correction, and then assembling. However, we did see a slight improvement for the Human gut data set in the number of single-contig MAGs when DeChat read correction was applied prior to nanoMDBG. This and the fact that the read error correction remains a resource-intensive and time-consuming task in existing long read assemblers, including state-of-the-art genomic assemblers such as hifiasm[31] and Verkko[32], motivates the development of more efficient correction methods. Our minimizer-space correction approach introduces a more efficient paradigm, which we anticipate will inspire future methods to address this problem and provide fast and scalable base-level correction.

To conclude, we have shown that nanoMDBG coupled to the latest ONT low error rate reads can generate comparable numbers of high-quality MAGs as the more expensive but higher accuracy HiFi PacBio reads at the same sequencing depth. This, we believe, will allow, in the

future, perhaps with additional refinements, for comprehensive genomic surveys of even the most complex microbiomes.

## Methods

Figure 1 summarizes the steps of nanoMDBG, which we describe in more detail below.

### Preliminaries

We start with a lexicon of some terms and concepts related to MDBGs, assembly and sequence alignment.

**Minimizer.** In order to scale up large datasets, our method uses only a fraction of the $k$-mers present in the reads. To select $k$-mers, we use the well-known concept of minimizers, more specifically, universal minimizers[14]. Given a hash function $f$ that maps $k$-mers to [0, $H$], we select a $k$-mer $m$ as a minimizer if $f(m) < dH$ where $d$ controls the fraction (or density) of selected $k$-mers.

**Minimizer-space read (mRead).** Prior to any treatment, reads are scanned and transformed into their ordered list of minimizers, which we call a mRead. We record the positions and orientations (forward/reverse) of minimizers in each read.

**Seed-and-chaining.** The seed-and-chaining strategy is a heuristic to quickly localize potential alignments between two sequences[26,33,34]. Read mapping tools usually rely on this technique prior to applying a more costly base-level alignment algorithm (hence the full technique is named "seed-chaining-extend"). Seed-and-chaining is also the core procedure of nanoMDBG (but the extended step will be different).

It starts by collecting a subset of $k$-mers (the seeds) from the sequences to compare. In our case, we use all universal minimizers in reads as seeds. The chaining step then finds exact seed matches between the sequences (i. e., anchors) and identifies sets of colinear anchors (the chains). For identifying the optimal chains, we use the same scoring function and banded dynamic programming algorithm as defined in skani[33].

**Minimizer-space alignment.** The seed-and-chaining procedure is usually a preliminary step prior to base-level alignment. However, here, the entire alignment step will operate in minimizer-space, and we will not go back to base-space. We refer to this as minimizer-space alignment. Sequence divergence can be directly estimated from the results of chaining as follows[35]:

$$e = \left(\frac{n}{m}\right)^{1/k} \tag{1}$$

where $n$ is the number of matching seeds, $m$ is the number of seeds in the query, and $k$ is the minimizer length.

To see the difference with base-level alignment, we explain here how alignment is performed with minimizers as units (see Fig. 1b) instead of bases. For a pair of mReads and matching minimizers (anchors), the other minimizers present between these anchors are examined to derive a sequence of matches, mismatches, insertions and deletions of minimizers. Such a sequence is the minimizer-space alignment. It is crucially different from standard alignment algorithms, which use minimizers for chaining and do not further try and align the other minimizers not involved in chains: they then align all bases, which is more computationally expensive. Here, we perform an alignment on all minimizers.

### Algorithmic challenges and motivations

At the core of metaMDBG is a variant of the de-Brujin graph tailored for long reads, the minimizer-space de-Brujin graph (MDBG). This approach considers only a small subset (around 0.5%) of k-mers from

each read, selected uniformly using a minimizer-based sampling technique. The minimizers are then linked into chains of size $k\prime$ and further extended using a de Bruijn graph-like framework. In order to exploit the full potential of long reads, metaMDBG creates long and specific chains ($k\prime \approx 100$). However, in the presence of sequencing errors, such long chains inevitably end up containing erroneous $k$-mers, which break contiguity (see ref. 14 for a detailed study of how sequencing errors in base-space propagate to minimizer-space). In this study, we aim to extend metaMDBG to support ONT data assembly through read correction, while preserving computational efficiency to enable the assembly of large metagenomes.

Self-correction of long reads typically relies on multiple sequence alignment. This approach involves selecting each read as a target for correction, recruiting and aligning similar reads to it, and identifying correct or erroneous bases through read pileup or partial order alignment (POA) graph. This correction procedure involves an initial all-vs-all read mapping and base-level multiple sequence alignment, which are both computationally expensive. Our goal is to make this process more efficient.

Our first ingredient to speed up read comparisons is to bypass base-level alignment, performing correction at the level of minimizers only. This approach mirrors the principles of base-level correction, with mReads aligned with a fast seed-and-chaining procedure instead, and minimizers stacked up to produce a consensus. The corrected mReads are then assembled in minimizer-space using metaMDBG, with base-level correction deferred to the final non-redundant contigs.

Despite this efficient approach, the all-vs-all step remains computationally demanding due to the high density of minimizers required for accurate read recruitment. To address this, we introduce a second ingredient, a two-step alignment strategy tailored for read correction, based on three key observations:

1. ONT read self-correction in minimizer-space can achieve high accuracy with moderate coverage (around 20×) of recruited reads.
2. For a given target read, recruiting candidate reads that are likely to belong to the same species and to the same genome location can be done quickly using the following heuristics. (1) Longer alignments are more likely to be correct than shorter alignments. (2) Alignment bounds can be estimated quickly using seed-and-chaining with low-density of minimizers.
3. Confirming that the candidate recruited reads are indeed location- and species-compatible with the target necessitates more effort, as it requires estimating sequence divergence. This step requires high-density minimizers, especially on shorter reads[33].

Our correction strategy, therefore, proceeds as follows. For each target read, we recruit the most similar reads in (sparse) minimizer-space (up to approximately 20x coverage) using seed-and-chaining with a very-low density of minimizers. We then remove divergent reads using a higher density of minimizers. This strategy restricts the more costly divergence analysis to a subset of reads, all read vs. approximately 20× coverage, rather than requiring it for all reads. The following section describes our correction strategy and heuristics in more detail.

### Read correction in minimizer-space

The following steps describe the novel correction step specifically (see Fig. 1c). In a nutshell, we first perform a fast all-vs-all read mapping using a low density of minimizers (0.5% by default). The output of this step is, for each read, the list of its best read matches (up to 20× reads). For a given target read to correct, we recompute its alignments using a higher density of minimizers (2.5% by default), and filtered recruited reads more accurately. We then pile up the minimizers using a variation graph and determine the most supported path as the consensus. We describe each of these steps more precisely in the following.

**Converting to minimizer-space.** We start by converting the reads into their ordered list of universal minimizers, which we call mReads. We extract two sets of mReads. One with a low-density of minimizers (0.5% by default), used to quickly perform all-vs-all read mapping, and another with a high-density of minimizers (2.5% by default), used to accurately correct mReads. We note that certain minimizers can be extremely abundant, which significantly impacts mapping performance. To address this, we filter out the top 0.0001 fraction of repetitive minimizers from both correction and assembly.

**Ultra-fast mRead recruitment using low-density minimizer-space alignment.** In this step, we perform all-vs-all read seed-and-chaining in order to find for each read its best matching reads that will be used for correction. In order to speed up this costly process, we use a low density of minimizers (0.5% by default). We use the following simple formula to rank alignments: score = number of minimizer matches − number of differences, where the number of differences is the sum of mismatches, deletions and insertions. For each read, we recruit up to 20x reads. Other alignments are discarded.

**Accurate high-density alignment filtering.** In this step, we perform seed-and-chaining alignment between the target mRead and the recruited mReads using their full minimizer representation (2.5% by default). Alignments with more than 4% divergence are discarded in order to discard reads that belong to other species while allowing for a certain error rate. Additionally, we discard alignments smaller than 1000 bps or with overhang longer than 2000 bps, in order to correctly recruit reads in genomic repeats.

**Piling up minimizer with minimizer-space variation graph from high-density alignments.** After accurate alignment filtering, we start the construction of a variation graph from the remaining alignment (see Fig. 3c). We initiate the graph with the minimizers of the target mRead. We then process sequentially each high-density alignment computed in the previous step. A node is added to the graph when a mismatch or an insertion occurs (see, respectively, node m7 and node m8 in Fig. 3c). We also record the support of each minimizer transition.

We note that this is a simpler approach than a partial order graph (POA), as alignments are performed only against the target mRead, rather than against a graph as in the case of POA. We found that the correction with this strategy was good enough for assembly. It has the advantages of being faster than the POA, and the resulting variation graph does not depend on the order in which sequences are added to the graph.

If a FASTQ file is provided as input, we weight edges by quality scores instead of raw transition counts. The quality score of a minimizer is defined as the minimum quality score among its constituent bases. Each edge in the graph is then weighted by the sum of the qualities of its source and destination minimizers.

**Generating consensus.** After all the alignments have been added to the variation graph, we extract the most supported path from the graph as a consensus using the following procedure. We begin by initializing a hash table $S$ that maps each node $v$ in the graph to its support score, with all values initially set to zero: $S[v] = 0$. Next, we perform a topological sort of the graph's nodes and iterate through them in the sorted order. For each node $v$, we identify the highest-supported incoming edge $e = (u, v)$ with weight $w_e$, and update its support score as $S[v] = S[u] + w_e$. The node $u$ is recorded as the parent of $v$ for path reconstruction. The consensus path is then constructed by identifying the node $v_{max}$ with the highest support score $S[v]$, and backtracking through its parent nodes until a node with no parent is reached.

## Assembling corrected minimizer-space reads

The corrected mReads are assembled using metaMDBG[9], with only minor modifications for the conversion of contigs from minimizer-space to base-space. A brief summary of the methods of MetaMDBG is provided here for completeness. MetaMDBG uses the minimizer de-Bruijn graph as a core structure. It assembles mReads using an efficient multi-k strategy in minimizer-space for handling uneven species coverage. At each iteration of the multi-k algorithm, an abundance-based algorithm termed "local progressive abundance filter" is applied to remove potential inter-genomic repeats, strain variability and complex error patterns. The resulting minimizer-space contigs (mContigs) are finally converted into their base-space representation. Previously, our method for reconstructing mContig sequences was designed for highly accurate reads. However, for ONT data, we developed a more adaptable approach. We map the raw reads (in minimizer representation) to the mContigs using the seed-and-chaining algorithm that we developed for our correction module. We then identify the longest alignments and extract the sequences from the raw reads that are spanned by the minimizers to reconstruct the mContig sequences. The final draft assembly is polished using a racon-like[36] approach.

## Modifications to metaMDBG to improve assembly quality

To enhance assembly quality, we adjusted some parameters and improved the polishing workflow. To reduce assembly errors, we configured the assembler to be less aggressive at the cost of fewer near-complete contigs compared to previous versions. Specifically, we increased the minimizer size from 13 to 15, and lowered the local abundance threshold used to filter out variability and inter-genomic repeats in the assembly graph from 0.5 to 0.25 (parameter $\beta$ in the original metaMDBG study). For example, for a species with 50x coverage, we now filter out its neighboring contigs in the graph with abundance ≤12.5 (50 * 0.25). We also addressed an issue in our racon-like polishing implementation that was a source of clipping events in the final assembly. The 500 bp polished contig windows could be too short due to a trimming step. We improved contig dereplication by enabling the base-level alignment mode of minimap2 during all-vs-all contig mapping. This mode was previously disabled to reduce memory usage, but doing so could result in missed overlaps and duplication. Finally, we added a second round of contig polishing to improve base-level accuracy.

## Metagenome sequencing data generation

Detailed protocols for DNA extraction and platform-specific library preparations are available at (https://www.protocols.io/view/soil-metagenome-pacbio-djga4jse.html and https://www.protocols.io/view/soil-metagenome-ont-dhrm3546.html). A summary of sample collection, DNA extraction and library preparations is provided here.

**Sample collection and storage.** All samples were collected and stored in Zymo DNA/RNA shield and left to incubate at 4 °C for 3 h prior to snap freezing and storage at −80 °C prior to DNA extraction.

Soil was collected using a sterile soil corer and passed through a sterile soil sieve to homogenize the sample. A total of 10g of homogenized soil sample was mixed with 100 ml of Zymo DNA/RNA shield and distributed into 1 ml aliquots containing 100mg/ml of soil.

Fecal samples were provided as part of an ongoing human study, "QIB colon model study". The study was approved by the local Quadram Institute Bioscience Human Research Governance committee (IFR01/2015) and by the London-Westminster Research Ethics Committee (15/LO/2169). The trial was registered at clinicaltrials.gov (NCT02653001). All participants provided signed informed consent prior to donating samples. The study was conducted in accordance with the Declaration of Helsinki.

Fecal samples are provided by volunteers of both genders, aged between 25 and 54. All participants declare to be in good health, have

no diagnosed chronic gastrointestinal health problems, and have not consumed antibiotics for at least six months prior to stool donation. Donors are not instructed to follow any diets or consume any specific foods, in accordance with the terms of the ethical approval.

A total of 10g of fresh fecal material was processed in a stomacher with a total of 100 ml of anaerobic PBS. Fecal slurry was then collected and distributed into 1 ml aliquots. Aliquots were centrifuged for 10 min at 10,000 × $g$ at Room temperature, and the supernatant was discarded. Samples were then homogenized with 1 ml of Zymo DNA/RNA shield and stored at − 80 °C prior to DNA extraction.

**DNA extraction.** DNA was extracted from all samples using the MPBio Fast DNA spin kit for soil with slight modifications from the manufacturer's protocol. Modifications include processing samples on ice, a reduced homogenization time of 2 × 10 s at 5.0 m/s with 5 min of cooling between runs. DNA was eluted first into 100 ml of DES elution buffer at 56 oC with the addition of 1 ml of RNaseA. Samples were then subjected to a 0.5×SPRI clean-up prior to both PacBio and ONT library preparations.

**PacBio library preparation and sequencing.** PacBio libraries were constructed using the SPK 2.0 kit and the overhang SMRT bell adapter. Samples were size-selected using a 3.7× dilute SPRI cleanup prior to sequencing. Libraries were sequenced using the PacBio Revio device and high-density flow cells and ran for a total of 24 h movie time with a hifi q score cut off at q20.

**Nanopore library preparation and sequencing.** Nanopore libraries were constructed using LSK-114 and processed using the long fragment buffer workflow. Libraries were sequenced on a P2 solo device using R 10.4.1 flow cells (Flo-Pro 114M). Samples were sequenced for a total of 96 h with a score cut-off at q10. Sequences were base-called and trimmed using the Dorado SUP V5 basecalling model in post-processing.

**Assembling data sets, mapping reads, and binning contigs**
We ran all assemblers with 32 CPU threads on a machine equipped with a 2.60 GHz Intel(R) Xeon(R) Gold 6132 CPU with 112 cores and 2038 GB of memory. All programs were installed via Bioconda. We ran metaMDBG with the option "-in-ont" for ONT data (to activate nanoMDBG method) and with the option "-in-hifi" for HiFi. We ran metaFlye with the options "-meta" and "-nano-hq" for Nanopore data sets and with the option "-pacbio-hifi" for HiFi data sets. We ran hifiasm-meta with default parameters. We only used the hifiasm-meta primary assembly of polished contigs (p_ctg.gfa), as adding alternate contigs reduced the overall MAG quality. We used the command "/usr/bin/time -v" to obtain wall-clock runtime and peak memory usage. All tools that were used and the complete command line instructions are available in Suplementary Table 1.

We mapped reads to contigs using "minimap2" with option "-x map-ont" for ONT reads and with option "-x map-hifi" for HiFi reads. We filtered out reads in which all of the alignments were shorter than 80% of their length, and we assigned each remaining read to a unique contig through its longest alignment (breaking ties arbitrarily). We performed contig binning using SemiBin2[4] (v.2.1.0), using the "single_easy_bin" command with options "-sequencing-type=long_-read", "-self-supervised" and a fixed seed (-random-seed 42) for reproducibility.

**Quality assessment of assemblies**
We used CheckM2 (v.1.0.1) to assess the quality of all MAGs. We used Barrnap (https://github.com/tseemann/barrnap), and Infernal[37] to predict, respectively, rRNA and tRNA genes from near-complete MAGs. We filtered out annotations with E-values over 0.01. We denoted near-complete MAG as RNA complete if they have at least one full-length

copy for all three types of rRNAs, and at least 18 full-length copies of tRNAs. We used geNomad (v.1.8.0) using the command "end-to-end" with options "-conservative" to identify strongly supported plasmids and viruses in each assembly. We used CheckV[19] (v.1.0.3) to assess the quality of viral contigs.

**Quality assessment of assemblies on the Zymo mock community**
We use MetaQUAST (v.5.2.0) with options "-unique-mapping" and "-reuse-combined-alignments" to assess assembly quality against mock reference genomes. Metrics reported include completeness, contiguity (auNGA), the number of mismatches and indels per 100 kbp, and the number of misassemblies. We also reported a "status" metric, defined as "circular" if a circular contig covers at least 99% of the reference genome, or "single-contig" if the contig is linear.

**Taxonomic classification of MAGs recovered from the ONT soil sample**
The phylogenetic tree of Fig. 2c was built using fasttree[38] from the output alignment of gtdbtk version 2.1.0[39] on near-complete quality MAGs of all three assemblers for the anaerobic digester dataset. Concurrent diversity coverage between the different assemblers was explored at different taxonomic levels from genus to domain. To do so, it is necessary to first address MAGs for which no annotation is available at a given taxonomic rank. A pair of unannotated MAGs may or may not share the same taxa. A first pass based on tree topology allows us to select neighboring MAGs as candidates for sharing the same unknown taxa. As a second step, we compute the Relative Evolutionary Distance using the R library Castor version 1.7.3[40]. Following guidelines from gtdb, we use their median RED values for each taxon in order to decide on grouping together unknown MAGs. We then find the best ancestor for each unknown MAG in terms of their RED being nearest to the corresponding taxa median RED. If they share the same best ancestor, we group them together; otherwise, we split them into distinct unknown taxa. Tree manipulation and representation is carried out using the library ggtree version 2.4.1[41], treeio version 1.14.3[42] and ggtreeExtra version 1.0.2[43].

**Identification of single-copy core genes (SCGs)**
In order to assess the species diversity of the assemblies and hence quantify our ability to capture this diversity as MAGs, single-copy core genes (SCGs) were identified from contigs. First, open reading frames (ORFs) were called using prodigal[44] (v2.6.3) with the option -p meta. A set of 36 different SCGs was annotated using RPS-BLAST (version 2.16.0)[45] using the pssm formatted COG database[46], which is made available by the CDD[47]. Annotation was performed with a best hit strategy, filtering out any hit with an e-value higher than 1e-10 and for which the alignment represents less than 50% of the subject length. Resulting hits were de-replicated at 97% using vsearch (2.29.1)[48] with option -cluster_smallmem. Species diversity was estimated as the median abundance across all SCGs.

A custom Python (version 3.10) script was used to link previously computed contig coverages, inclusion of contig in a MAG, related MAG quality and SCGs. Coverage of SCGs from the same cluster was summed, and when multiple MAG qualities could be found, the highest one was taken. This resulted in a table stating for each SCG cluster, its coverage and if it was found included in a MAG as well as related quality. This table was plotted using the ggplot2[49] package from R (version 4.3.3) to give Fig. 3.

**Identification of errors in assemblies**
We used CheckM2 (v.1.0.1) to assess contamination in contigs longer than 1 Mb. Clipping events and zero-coverage regions were identified using the anvi-script-find-misassembly program from the Anvi'o platform[25], following the protocol described in ref. 23. This script relies on read mapping to the assemblies using minimap2 with the options

`-a -p 1 --secondary-seq`. It extracts clipping events (hard or soft clipping) directly from the CIGAR strings of the resulting BAM file. We reported clipping events within the assemblies that were supported by at least 10 reads, with 100% of those reads exhibiting clipping. We used IDEEL[24] with default parameters on contigs longer than 5 kb and with coverage over 10x to quantify the fraction of predicted proteins that were ≥90% the length of their best-matching known protein.

## Data availability

The sequence data generated in this study have been deposited in the European Nucleotide Archive as the BioProject PRJEB88618. The individual accession numbers of all sequences used are: ERR15316007: Zymo ONT; ERR15285694: Human gut ONT; ERR15289757: Soil ONT; ERR15289675: Human gut HiFi; ERR15289804: Soil HiFi. Zymo mock reference genomes are available at https://s3.amazonaws.com/zymo-files/BioPool/D6331.refseq.zip. The ONT Zymo Fecal Reference data set is available at https://epi2me.nanoporetech.com/lc2024-datasets/. The HiFi Zymo Fecal Reference data set is available at https://www.pacb.com/connect/datasets/#metagenomics-datasets. Source data are provided with this paper.

## Code availability

We implemented the nanoMDBG method in the metaMDBG software (https://github.com/GaetanBenoitDev/metaMDBG). The nanopore mode is activated using the input parameter (`--in-ont`), and the original PacBio HiFi mode using the parameter (`--in-hifi`). The analysis scripts used in this study to compare assemblers are available at https://github.com/GaetanBenoitDev/NanoMDBG_Manuscript.

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

## Acknowledgements

C.Q. and S.R. acknowledge the support of the Biotechnology and Biological Sciences Research Council (BBSRC), part of UK Research and Innovation; Earlham Institute Strategic Program (ISP) Grant (Decoding Biodiversity) BBX011089/1 and its constituent work package BBS/E/ER/230002C; the Core Strategic Program Grant (Genomes to Food Security) BB/CSP1720/1 and its constituent work packages BBS/E/T/000PR9818 and BBS/E/T/000PR9817; and the Core Capability Grant BB/CCG2220/1. C.Q. and R.J. acknowledge the QIB Food Microbiome and Health ISP BB/X011054/1 and its constituent project BBS/E/F/000PR13631. The authors gratefully acknowledge the support of the QIB Colon Model Facility, which was funded by the BBSRC Core Capability Grant BB/CCG2260/1. R.C. was supported by ANR grants ANR-22-CE45-0007, ANR-19-CE45-0008, PIA/ANR16-CONV-0005, ANR-19-P3IA-0001, ANR-21-CE46-0012-03, and Horizon Europe grants No. 872539, 956229, 101047160 and 101088572 (ERC IndexThePlanet, also supporting G.B.). We acknowledge the assistance of Dr. Susheel Bhanu Busi (CEH, Wallingford) in organizing the soil sampling.

## Author contributions

G.B. devised and implemented the approach and performed analysis with assistance from S.R., R.J., and G.A. prepared DNA extracts for sequencing and constructed libraries. T.G. collected soil samples. G.B., R.C., and C.Q. conceived the study and supervised and coordinated the work. All authors wrote, reviewed, edited and approved the manuscript.

## Competing interests

The authors declare no competing interests.
