## [Transparent Peer Review file · Nature Communications]

High-quality metagenome assembly from nanopore reads with nanoMDBG

Corresponding Author: Dr Christopher Quince

Version 0:

Reviewer comments:

Reviewer #1

(Remarks to the Author)

This manuscript introduces nanoMDBG, a long-read metagenome assembly algorithm for Oxford Nanopore sequencing data. The authors benchmark it on a range of datasets and show that it can produce more complete assemblies than other long-read metagenome assemblers while using fewer computational resources. I have only minor comments, which are listed below.

In the Data availability section, it states: "The sequencing read data generated for this study are available at ENA bio project PRJEB88618; accession numbers are given in Table 1." However, Table 1 does not currently include any accessions. While I could see that project on ENA, I was unable to find any associated read sets. Please ensure all read sets are publicly available and add their accessions to Table 1.

The topic of strain mixtures and their impact on metagenome assembly is only mentioned briefly: in the Zymo mock community results ("No assembler could correctly resolve all of the E. coli strain diversity...") and in the discussion of the soil metagenome ("...it may be strain variation..."). I think this topic merits further discussion. For example, when faced with a strain mixture, should a long-read metagenome assembler attempt to reconstruct each strain separately, even if that leads to fragmentation (as metaFlye may be doing)? Or should it favour longer contigs at the cost of collapsing diversity (as nanoMDBG may be doing)? Since strain mixtures are very common in metagenomes, a paragraph exploring this trade-off would be a valuable addition to the Discussion.

When introducing nanoMDBG, it should be made clearer that it is a new algorithm/mode within metaMDBG, not a standalone tool. For example, the phrase "introduce an application, nanoMDBG" suggests it is separate. This distinction is clarified only in the Code availability section, but it would help readers to explain this earlier in the manuscript.

The sentence "The initial step in metagenomic analysis is the assembly of sequencing reads into longer contiguous fragments or contigs" may overstate the generality of assembly in metagenomics. Many analyses use read-based taxonomic profiling without performing assembly. A more accurate phrasing might be: "The initial step in many metagenomic analyses is..."

The minimap2 preset for HiFi reads is given as "-map-hifi", but it should be "-x map-hifi".

Throughout the manuscript, command-line options are printed with a single dash instead of a double hyphen (--), possibly due to Latex formatting. Consider using a monospace font (e.g. `\texttt{--in-ont}`) to preserve formatting.

The Code availability section states that nanoMDBG is invoked using `--in-nano`, but it should be `--in-ont`.

(Remarks on code availability)

While I have not attempted to reproduce the specific results presented in the manuscript, I have run metaMDBG with the new nanoMDBG algorithm on my own datasets, including both metagenomes and isolate genomes, and it worked well.

Installation was straightforward using conda. The metaMDBG GitHub repository includes a README that covers the basics (installation, usage and output) but it is somewhat brief and contains typos. The documentation would benefit from additional

detail and a link to a small demo dataset to help new users get started. Also, the Advanced usage and Low-memory contig polisher sections are currently empty, so the README seems to be incomplete.

Reviewer #2

(Remarks to the Author)

The authors propose an extension of their previous minimizer space based metagenomic assembly (metaMDBG) to less accurate Nanopore reads, called nanoMDBG. The core improvement (and novelty) is an efficient error correction method that operates in minimizer space. This is done through a two step method via seed-chaining: For each read, first a lower density of minimizers are used to detect top 20 "similar" reads that are from the same species and same genomic region. Then, all minimizers on the selected reads are used for multiple sequence alignment on minimizer space to construct a variation graph and derive the consensus sequence as the most supported path in this graph. Rest of the assembly process is similar to metaMDBG, where a minimizer space DBG is constructed and progressive abundance filter (LPAF) is iteratively applied to suppress variance (either due to strain variation or errors). nanoMDBG with and without error correction is benchmarked against metaFlye and hifiasm-meta (in the case of HiFi reads). Authors also benchmark nanoMDBG and other assemblers after applying two base-level read correction methods. The data used for this benchmarking analysis contains mock and real datasets from different environments. nanoMDBG shows improvements in terms of number of single contig, near-complete and high quality MAGs recovered.

Overall, the manuscript is an important contribution as it makes a substantial improvement over the state-of-the-art methods for nanopore metagenomic assembly, and has potential to become a popular tool. We have a few specific requests:

1. Conceptually, it is currently unclear how exactly metaMDBG (assembly algorithm) benefits from corrected reads. What is the exact issue if metaMDBG is applied to uncorrected reads? If raw reads can be aligned during the error correction stage, why couldn't the same alignment be used to construct the assembly graph? Although the answers might be obvious for an assembly expert, the paper might benefit from making these points explicit.
2. What is the exact algorithm for generating consensus from the variant graph? Greedily selecting the most supported path at every junction may not yield the most supported overall path.
3. This sentence is a bit confusing: "Such a sequence is the minimizer-space alignment. It is crucially different from standard alignment algorithms, which use minimizers but only focus on anchors and do not further try and align the other minimizers present between the anchors, they align the bases, which is more computationally expensive". Do authors imply that pairwise minimizer alignment is different from minimizer chaining? If yes - how exactly? In what case an optimal (matched) minimizers chain will not be equivalent to an optimal pairwise alignment?
4. Benchmarking in terms of MAG recovery is a good strategy for real datasets, but it does not specifically address misassemblies, contiguity and sequence-level quality. For datasets where reference is available (i.e., Zymo Mock), metaQUAST could be used to generate these metrics.
5. For real datasets, an indirect measure of misassemblies could be the number of MAGs with high completion, but elevated contamination (e.g. 10%+) - perhaps this metric could be added for real datasets comparison?
6. Base-level consensus quality has been a concern for nanopore assemblies. For real datasets (when references are not available), one could use tools like IDEEL (<https://github.com/mw55309/ideel>) to indirectly estimate the number of frameshift errors.
7. It would be interesting to include species similarity analysis in different datasets. Strain similarity can also explain the differences in complexity and assembly quality for the datasets used in the paper.

Minor:

8. In HiFi vs Nanopore comparison (Figure 4), Is there any explanation as to why for Human gut, HiFi performs better with increasing coverage but Nanopore and HiFi perform similarly for Zymo Fecal and Soil samples. This analysis is not very conclusive overall.
9. Does the error rate in minimizer space correspond to base-level error rate for different reads? Although it's expected - since the primary focus of the paper is error correction, this analysis might be important.
10. Zymo mock results description is inconsistent: "In this case, nanoMDBG and metaFlye both obtained five circularized genomes and metaMDBG obtained four. [...] In contrast, metaFlye produced only fragmented genomes." Seems to be a contradiction with metaFlye results.
11. Runtime and peak memory values in the supplementary data do not seem consistent. For ONT Human Gut (50Gb) why does metaMDBG take longer than nanoMDBG where nanoMDBG applies error correction on top of the general assembly steps that metaMDBG follows? For ONT Zymo Fecal Reference (200Gb), it's surprising to see metaFlye the fastest? These results require some more explanation.

(Remarks on code availability)

The repository is in a good shape.

Reviewer #3

(Remarks to the Author)

(Remarks on code availability)

Reviewer #4

(Remarks to the Author)

Benoit et al present nanoMDBG, a substantial modification of the metaMDBG algorithm to enable it to assemble ONT readsets. The benchmarks as presented are very impressive, and while I quibble about such strong self-endorsement such as suggesting the algorithm “could prove transformative to metagenomics research”, I do think nanoMDBG has the potential to be the go-to assembler for ONT reads. These are increasingly common in metagenomics, and therefore nanoMDBG appears to be an extremely useful resource for the community. Initial application of the tool to my own datasets confirms the minimal RAM requirements (and ease of installation through bioconda) even for large and complex communities (assembly and binning is on-going so I cannot comment on those aspects).

My two main criticisms of the manuscript lie with benchmarking the accuracy of the contigs produced. I would argue this is imperative for metagenome assemblers to present, but it is especially pertinent given a recent prepublication which finds several error modes find their way into metaMDBG assemblies (<https://www.biorxiv.org/content/10.1101/2025.04.22.649783v2>). While that manuscript is not passed peer review, it highlights the paucity of contig accuracy benchmarks from this nanoMDBG manuscript. The tools developed in that preprint may be helpful for analysis here.

Beyond overall benchmarks of per-base accuracy, chimeras, and other error modes, there is another issue with the approach which should be at least discussed in the manuscript, or even better analysed in detail. The issue is around the strain-specificity of the contigs produced. Conceptually, if a read from a non-dominant strain is “corrected” by the method proposed, then it may be “corrected” to the genotype of the dominant strain/genotype if that genotype shares regions which are sufficiently similar, especially since only a 20x coverage limit is applied in the initial step, precluding some analysis which might differentiate sequencing error from real but low-abundance variation. On the other hand, if that were to happen then it might artificially inflate the coverage of the dominant strain, aiding in its eventual genome recovery when it might otherwise be too sparsely covered. Perhaps these considerations also apply to other assemblers, but I suggest that they are non-trivial enough that they should be at least mentioned for reviewer context.

Minor comments.

Please include line numbers in future versions of the manuscript to assist reviewers — it greatly aids clarity and precision in feedback.

In general referencing in the introduction could do with some more care. For instance, the statement “MAGs constructed from short reads frequently suffer from incompleteness, contamination and high Fragmentation.” is unreferenced, and the first mention of metaMDBG is not referenced, etc.

“ntigs into metagenome-assembled genomes (MAGs) [1],” - this [1] should be replaced by reference(s) which better represent the state of the field, rather than a manuscript by the same corresponding author.

“In contrast, metaFlye produced only fragmented Genomes” - this section could be written more clearly. It isn't clear which genomes are being referred to here, for instance. Perhaps it would read better if the E. coli strains and low abundance species were discussed separately, rather than everything at once. It would also be clearer if the true abundance table was given, so that the precise meaning of “low abundance” is immediately available to the reader, rather than asking them to chase this down through the Zymo website (an option which may also be not available in the long term).

“Significantly” - word should be replaced as no statistical analysis is present (nor should there be).

Capitalisation of bioninformatic tool names should be made more carefully. For instance the first C of CheckM2 is capitalised (there were several variants in the ms), and the tool name is CheckM2, not CheckM.

The words “near complete” and “complete” are used to define categories of MAG reconstruction accuracy. However, these words have precise meaning as defined in the MIMAG standard, and they should not be changed to suit a particular study. They also incorporate genome characters beyond CheckM2-based completeness and contamination statistics. For instance, requirements to include rRNA operons. One option might be to use one threshold of contamination (say 5%), and then the

categories names can reflect different levels of completeness. This way more subtle trends than high quality vs medium quality can be showcased, while not confusing the precise categorisations set up in the MIMAG standard.

“Not every read will be represented in the assembly, .. “ - I did not understand the logic for excluding unassembled reads from this percentage. I suggest rephrasing or removing this interpretation.

“We note that we ..” over-use of “we” here. There is an argument also that is true more generally in the manuscript, but perhaps that is a matter of taste.

“Varying sequencing depths”: On first read I worried about non-equal read quantities being used, which might bias the results. Further reading allayed my concerns on this, but perhaps this sentence could be clearer on that point up front.

“The improved MAG recovery by nanoMDBG translates into a more representative picture of microbial diversity at all levels of evolutionary divergence.” While this is likely true, it cannot be concluded so confidently, because the true diversity remains unknown.

“Specific” vs “shared” - are these the same meaning? If so only 1 should be used to avoid confusion.

“Gut” or “human gut”? Only one descriptor should be used, to avoid confusion.

“Proved essential”. It may not have been essential, even if the manuscript shows it helps assembly quality. Suggest rephrase.

“the latest ONT sequencing technology now produces results comparable to PacBio HiFi sequencing at equivalent sequencing depth, despite the remaining differences in raw read accuracy.” - this is true only if the per-base accuracy of the contigs is comparable, which has not been established.

“read
Metagenomics” - error

“adapt our algorithm to a streaming assembler, “ - grammatical mistake. This sentence is also too long.

“The most similar reads (up to approximately 20x coverage)”. This should be qualified to mean that they are the most similar in (sparse) minimizer space, which is not the same thing.

“Filtired” - grammatical mistake.

“And”: should not start a sentence.

“Perform against” - grammar mistake.

“Generating consensus” - this section needs more detail.

“At each multi-k” - is this multi-k, or just each k?

A description of the hardware architecture used should be given, if walltime is the main output of the benchmark. How was it compiled? With optimisations for the specific hardware or a general-purpose build for e.g. conda?

“-map-hifi” is not an option of minimap2

“checkV” - which version?

“Assessment of completeness..” Does this section refer only to the Zymo community? If so, it would be clearer to say that directly.

Emdashes - in several places it appears “--” has been replaced by an emdash, which is incorrect if they refer to command line options.

Figure 4 “showcase” - Consider replacing ‘showcase’ with a more precise verb like ‘illustrates’ or ‘demonstrates’ to maintain academic tone.

“The highest sequencing depth available” - this was unclear to me - 50, 200 and 250G respectively? Or the full depth available for each method for each assembler?

Figure S3 “assemblers” should be singular. Y-axis Virus should be plural. Gbps - the s is not needed.

Reviewed by Ben Woodcroft

(Remarks on code availability)

I was able to run the code without issue.

Version 1:

Reviewer comments:

Reviewer #1

(Remarks to the Author)

The authors have addressed the comments from my previous review of this manuscript.

One minor typo in the Introduction: 'The initial step in many metagenomic analysis' should be 'The initial step in many metagenomic analyses'.

(Remarks on code availability)

Reviewer #2

(Remarks to the Author)

The authors addressed our comments and made the appropriate changes to the text where necessary, including additional discussions, explanations and analyses. We have only a couple minor comments below that do not require re-review.

Proportion of read correction time for the same dataset in lines 82 and 322 (10% and 6%) are not consistent.

We appreciated the expanded discussion on the issue of strain assemblies. The authors mentioned that metaFlye attempts to reconstruct strain genomes, but the more recent work from this group (<https://pubmed.ncbi.nlm.nih.gov/39327484/>) might be more relevant in this context.

(Remarks on code availability)

Reviewer #3

(Remarks to the Author)

(Remarks on code availability)

Reviewer #4

(Remarks to the Author)

In responding to these reviews, the authors have made substantial improvements that greatly benefit the manuscript.

In the IDEEL comparison, I am curious what the effect of a more fragmented assembly might have on the results. In particular, if many contigs are short, then ORFs on the ends of contigs might be substantial in number. I note that the `-c` parameter of prodigal is not used in IDEEL (maybe it should be, though of course that is not the tool being assessed here). The authors might choose to restrict the IDEEL analysis to contigs over a certain length, or even compare between different assemblers just those contigs in length bins e.g. 5-10kb, 2-5kb, etc. It is particularly a problem if one assembler uses a different minimum output contig length threshold - I have not looked into this.

The methods for the next section appear to be incomplete. For instance, it was not explained how clipping counts were determined.

273: It was not clear from the text what datasets were used for the first long contigs benchmark. I suppose it is the union of all contigs from all datasets assembled separately?

228-9: Should be past tense.

Reviewed by Ben Woodcroft

(Remarks on code availability)

RESPONSE TO REVIEWERS' COMMENTS

Reviewer #1 (Remarks to the Author):

This manuscript introduces nanoMDBG, a long-read metagenome assembly algorithm for Oxford Nanopore sequencing data. The authors benchmark it on a range of datasets and show that it can produce more complete assemblies than other long-read metagenome assemblers while using fewer computational resources. I have only minor comments, which are listed below.

In the Data availability section, it states: "The sequencing read data generated for this study are available at ENA bio project PRJEB88618; accession numbers are given in Table 1." However, Table 1 does not currently include any accessions. While I could see that project on ENA, I was unable to find any associated read sets. Please ensure all read sets are publicly available and add their accessions to Table 1.

We have added the accessions for the newly sequenced datasets in Table 1. These are all now publicly available for download.

The topic of strain mixtures and their impact on metagenome assembly is only mentioned briefly: in the Zymo mock community results ("No assembler could correctly resolve all of the E. coli strain diversity...") and in the discussion of the soil metagenome ("...it may be strain variation..."). I think this topic merits further discussion. For example, when faced with a strain mixture, should a long-read metagenome assembler attempt to reconstruct each strain separately, even if that leads to fragmentation (as metaFlye may be doing)? Or should it favour longer contigs at the cost of collapsing diversity (as nanoMDBG may be doing)? Since strain mixtures are very common in metagenomes, a paragraph exploring this trade-off would be a valuable addition to the Discussion.

We agree with the reviewer that this interesting topic deserves further discussion. We have added both additional sentences to the Introduction (lines 59-62) and a paragraph to the Discussion:

“355-360: In general, the philosophy behind nanoMDBG is to try to obtain as many species or sub-species consensus genomes as possible. This is perhaps different to metaFlye which may be attempting to reconstruct each strain separately...”

When introducing nanoMDBG, it should be made clearer that it is a new algorithm/mode within metaMDBG, not a standalone tool. For example, the phrase "introduce an application, nanoMDBG" suggests it is separate. This distinction is clarified only in the Code availability section, but it would help readers to explain this earlier in the manuscript.

We have modified the following sentences in the introduction, in the overview section and in the discussion:

"We introduce nanoMDBG, a novel algorithm integrated into the metaMDBG software, which revisits the concept of read correction in minimizer-space to address errors in ONT data."

"We present nanoMDBG, a metagenome assembly method within metaMDBG, designed for Oxford Nanopore Technologies (ONT) long-read data."

"We have introduced nanoMDBG, a new metagenome assembly method within metaMDBG designed for ONT long reads."

The sentence "The initial step in metagenomic analysis is the assembly of sequencing reads into longer contiguous fragments or contigs" may overstate the generality of assembly in metagenomics. Many analyses use read-based taxonomic profiling without performing assembly. A more accurate phrasing might be: "The initial step in many metagenomic analyses is..."

This has been changed according to the suggestion of the reviewer (Line 31).

The minimap2 preset for HiFi reads is given as "-map-hifi", but it should be "-x map-hifi".

Throughout the manuscript, command-line options are printed with a single dash instead of a double hyphen (--), possibly due to Latex formatting. Consider using a monospace font (e.g. `\texttt{--in-ont}`) to preserve formatting.

The Code availability section states that nanoMDBG is invoked using `--in-nano`, but it should be `--in-ont`.

We have fixed the above formatting and issues with precise parameter specification.

Reviewer #1 (Remarks on code availability):

While I have not attempted to reproduce the specific results presented in the manuscript, I have run metaMDBG with the new nanoMDBG algorithm on my own datasets, including both metagenomes and isolate genomes, and it worked well.

Installation was straightforward using conda. The metaMDBG GitHub repository includes a README that covers the basics (installation, usage and output) but it is somewhat brief and contains typos. The documentation would benefit from additional detail and a link to a small demo dataset to help new users get started. Also, the Advanced usage and Low-memory contig polisher sections are currently empty, so the README seems to be incomplete.

We have updated the usage instructions on the GitHub README, and revised typos, and completed the Advanced Usage section.

Reviewer #2 (Remarks to the Author):

...

Overall, the manuscript is an important contribution as it makes a substantial improvement over the state-of-the-art methods for nanopore metagenomic assembly, and has potential to become a popular tool. We have a few specific requests:

1. Conceptually, it is currently unclear how exactly metaMDBG (assembly algorithm) benefits from corrected reads. What is the exact issue if metaMDGB is applied to uncorrected reads? If raw reads can be aligned during the error correction stage, why couldn't the same alignment be used to construct the assembly graph? Although the answers might be obvious for an assembly expert, the paper might benefit from making these points explicit.

MetaMDBG builds its assembly graph using a de Bruijn graph-like structure in minimizer-space, based purely on exact k-mer matches. Unlike string-graph-based assemblers (e.g. hifiasm-meta) that rely on full base-level read alignments, our assembly procedure does not perform any form of alignment. This exact matching strategy allows for extremely fast and memory-efficient graph construction. However, it is also highly sensitive to sequencing errors. A single base error can break a k-mer match and disrupt graph connectivity, much like in classical de Bruijn graph assemblers. While our error correction strategy

does rely on alignments, these are performed in sparse minimizer-space and are much less computationally intensive than full-read base-level alignments.

The fourth paragraph of the Introduction already discusses these limitations but we now present them more explicitly in the first paragraph of the "Algorithmic Challenges and Motivations" as follows:

“Lines 412-420 : At the core of metaMDBG is a variant of the de-Brujin graph tailored for long reads, the minimizer-space de-Brujin graph (MDBG). This approach considers only a small subset (around 0.5%) of k-mers from each read, selected uniformly using a minimizer-based sampling technique. The minimizers are then linked into chains of size k' and further extended using a de Bruijn graph-like framework. In order to exploit the full potential of long reads, metaMDBG creates long and specific chains ($k' \approx 100$). However, in the presence of sequencing errors, such long chains inevitably end up containing erroneous k-mers which break contiguity (see [10] for a detailed study of how sequencing errors in base-space propagate to minimizer-space). In this study, we aim to extend metaMDBG to support ONT data assembly through read correction, while preserving computational efficiency to enable the assembly of large metagenomes.”

2. What is the exact algorithm for generating consensus from the variant graph? Greedily selecting the most supported path at every junction may not yield the most supported overall path.

We do not greedily select the most supported edge but do indeed find the most supported overall path which is straightforward given a directed acyclic graph. We have added a detailed description of the method used to extract the most supported path in this section.

“Lines 485-492: After all alignments have been added to the variation graph. We extract the most supported path from the graph as a consensus using the following procedure. We begin by initializing a hash table S that maps each node v in the graph to its support score, with all values initially set to zero: $S[v] = 0$. Next, we perform a topological sort of the graph's nodes and iterate through them in the sorted order. For each node v , we identify the highest-supported incoming edge $e = (u, v)$ with weight w_e , and update its support score as $S[v] = S[u] + w_e$. The node u is recorded as the parent of v for path reconstruction. The consensus path is then constructed by identifying

the node v_{max} with the highest support score $S[v]$, and backtracking through its parent nodes until a node with no parent is reached.”

3. This sentence is a bit confusing: “Such a sequence is the minimizer-space alignment. It is crucially different from standard alignment algorithms, which use minimizers but only focus on anchors and do not further try and align the other minimizers present between the anchors, they align the bases, which is more computationally expensive“. Do authors imply that pairwise minimizer alignment is different from minimizer chaining? If yes - how exactly? In what case an optimal (matched) minimizers chain will not be equivalent to an optimal pairwise alignment?

Yes, our minimizer alignment is different from minimizer chaining. The key difference is that all minimizers are sought to be aligned, whereas in chaining, a colinear chain is sought, possibly discarding some minimizers, thus it is not an alignment. We have revised the paragraph as follows:

“Lines 407-410 : Such a sequence is the minimizer-space alignment. It is crucially different from standard alignment algorithms, which use minimizers for chaining and do not further try and align the other minimizers not involved in chains: they then align all bases, which is more computationally expensive. Here, we perform an alignment on all minimizers.”

4. Benchmarking in terms of MAG recovery is a good strategy for real datasets, but it does not specifically address misassemblies, contiguity and sequence-level quality. For datasets where reference is available (i.e., Zymo Mock), metaQUAST could be used to generate these metrics.

This is a good suggestion. We used MetaQUAST between references and mapped contigs to measure completeness, contiguity (auNGA), the number of mismatches and indels per 100~kbp, and the number of misassemblies. These results have been added as a new paragraph at the start of the section ‘Evaluation of bacterial genome reconstruction.’ We have also updated the methods in the “Quality assessment of assemblies on the Zymo mock community” section accordingly.

5. For real datasets, an indirect measure of misassemblies could be the number of MAGs with high completion, but elevated contamination (e.g. 10%+) - perhaps this metric could be added for real datasets comparison?

This is also a good suggestion, contamination is a useful way to assess real data sets, but we felt that contamination at the MAG level could be biased, as modern binning tools such as SemiBin2 use SCGs during the clustering process. Since assemblers do not optimize for SCGs, we chose instead to evaluate contamination at the contig level. Specifically, we examined long contigs (>1 Mb) and identified those with over 5% contamination, treating these as chimeric. The results of this analysis have been included in the new section 'Evaluation of metagenomics assembly errors' of the revised manuscript.

6. Base-level consensus quality has been a concern for nanopore assemblies. For real datasets (when references are not available), one could use tools like IDEEL (<https://github.com/mw55309/ideel>) to indirectly estimate the number of frameshift errors.

As suggested by the reviewer we applied IDEEL to contigs with coverage greater than 10x in order to assess the importance of frameshift errors and hence indirectly the base-level consensus quality of each assembler. The results of this analysis have been included in the revised manuscript (Figure 5d). Interestingly, for the R10.4 ONT sequences used here there is no noticeable difference in quality between HiFi PacBio and ONT except in the low coverage Soil data sets. Soil is also where we see an advantage for nanoMDBG over metaFlye in terms of a higher rate of predicted proteins aligning to the database.

7. It would be interesting to include species similarity analysis in different datasets. Strain similarity can also explain the differences in complexity and assembly quality for the datasets used in the paper.

We agree that potentially differences in assembly quality may derive from levels of strain variation. However, without actually resolving strains it is hard to quantify this. Our experience is that simply looking at frequencies of variants on contigs is not informative perhaps because of challenges in variant calling for metagenomics data. So whilst we have added some discussion of strain resolution to the Discussion section of the manuscript we decided not to add this additional analysis.

Minor:

8. In Hifi vs Nanopore comparison (Figure 4), Is there any explanation as to why for Human gut, Hifi performs better with increasing coverage but Nanopore and Hifi perform similarly for Zymo Fecal and Soil samples. This

analysis is not very conclusive overall.

The results in Figure 4 have changed somewhat as the result of more stringent error correction in nano/metaMDBG. Now we do see an advantage for HiFi in soil in terms of near-complete contigs. The better overall relative performance of HiFi on the gut data sets at high depth remains though but we have no obvious explanation for this.

9. Does the error rate in minimizer space correspond to base-level error rate for different reads? Although it's expected - since the primary focus of the paper is error correction, this analysis might be important.

This relationship has been thoroughly analyzed in the original paper on the minimizer de Bruijn graph (<https://www.sciencedirect.com/science/article/pii/S240547122100332X>, see section "How sequencing errors in base-space propagate to minimizer-space"). Base-level errors have a pronounced impact in minimizer-space: for example, a base-space error rate of 5% results in a 65.1% error rate in minimizer-space, while even a low base-space error rate of 0.1% leads to a 2.3% error rate in minimizer-space.

10. Zymo mock results description is inconsistent: "In this case, nanoMDBG and metaFlye both obtained five circularized genomes and metaMDBG obtained four. [...] In contrast, metaFlye produced only fragmented genomes." Seems to be a contradiction with metaFlye results.

This was an error that has been corrected, and in fact this whole section has been rewritten as a result of including the MetaQUAST results (see above).

11. Runtime and peak memory values in the supplementary data do not seem consistent. For ONT Human Gut (50Gb) why does metaMDBG take longer than nanoMDBG where nanoMDBG applies error correction on top of the general assembly steps that metaMDBG follows? For ONT Zymo Fecal Reference (200Gb), it's surprising to see metaFlye the fastest? These results require some more explanation.

The original version of MetaMDBG (v1.1) could be very slow on uncorrected data, because errors create massive and complex assembly graphs, requiring a longer cleaning process (particularly the superbubble detection step). However, in the new 1.2 version (described above), the cleaning implementation has been parallelized so this is now less of an issue.

MetaFlye can still be faster in some instances, for instance the Zymo Fecal Reference is a dataset with low species complexity but very high coverage which is the worst case for an algorithm that relies on all-vs-all read mapping as nanoMDBG does in the error correction step.

Reviewer #4 (Remarks to the Author):

Benoit et al present nanoMDBG, a substantial modification of the metaMDBG algorithm to enable it to assemble ONT readsets. The benchmarks as presented are very impressive, and while I quibble about such strong self-endorsement such as suggesting the algorithm “could prove transformative to metagenomics research”, I do think nanoMDBG has the potential to be the go-to assembler for ONT reads. These are increasingly common in metagenomics, and therefore nanoMDBG appears to be an extremely useful resource for the community. Initial application of the tool to my own datasets confirms the minimal RAM requirements (and ease of installation through bioconda) even for large and complex communities (assembly and binning is on-going so I cannot comment on those aspects).

My two main criticisms of the manuscript lie with benchmarking the accuracy of the contigs produced. I would argue this is imperative for metagenome assemblers to present, but it is especially pertinent given a recent prepublication which finds several error modes find their way into metaMDBG assemblies (<https://www.biorxiv.org/content/10.1101/2025.04.22.649783v2>). While that manuscript is not passed peer review, it highlights the paucity of contig accuracy benchmarks from this nanoMDBG manuscript. The tools developed in that preprint may be helpful for analysis here.

As we mention above, we have in response to points 5 and 6 raised by Reviewer 2 added a whole new section ‘**Evaluation of metagenomics assembly errors**’ which is a comprehensive evaluation of likely artefacts and assembly errors. This includes the results of the tools referred to by the reviewer from <https://www.biorxiv.org/content/10.1101/2025.04.22.649783v2> but also single contig contamination and the usage of IDEEL to detect frameshift errors as recommended by Reviewer 2. We thank both reviewers for their suggestions. In addition, as we explain in our preamble to these responses, the error rates for the meta/nanoMDBG v1.1 that was used in the initial version of this manuscript were relatively high compared to the other assemblers. This inspired the changes described in the new Methods subsection ‘**Modifications to metaMDBG to improve assembly quality**’ corresponding to the current v1.2.

Following these changes the metaMDBG error rates are comparable to the other state-of-the-art assemblers. So again we thank the reviewers for their suggestions which resulted in improvements to our underlying algorithms.

Beyond overall benchmarks of per-base accuracy, chimeras, and other error modes, there is another issue with the approach which should be at least discussed in the manuscript, or even better analysed in detail. The issue is around the strain-specificity of the contigs produced. Conceptually, if a read from a non-dominant strain is “corrected” by the method proposed, then it may be “corrected” to the genotype of the dominant strain/genotype if that genotype shares regions which are sufficiently similar, especially since only a 20x coverage limit is applied in the initial step, precluding some analysis which might differentiate sequencing error from real but low-abundance variation. On the other hand, if that were to happen then it might artificially inflate the coverage of the dominant strain, aiding in its eventual genome recovery when it might otherwise be too sparsely covered. Perhaps these considerations also apply to other assemblers, but I suggest that they are non-trivial enough that they should be at least mentioned for reviewer context.

Our assembler is designed to produce contigs that are consensus genomes at the species or at best sub-species level. This is implicit in our use of a relatively low density (0.5%) of short minimisers (kmer length 13). Strain-variation between these minimisers will be ignored. In our experience this limits resolution to about 1% average nucleotide identity (ANI). Consequently, the fact that the ONT error correction step may remove low-coverage variants as the reviewer suggests is not an issue. Since we are not attempting to assemble these strains in any case. That metaMDBG is not attempting strain resolution is an important point and in response to Reviewer 1 we have added a new paragraph to the Discussion “**In general, the philosophy behind nanoMDBG is to try to obtain as many species or sub-species consensus genomes as possible....**” which should help clarify this for readers.

Minor comments.

Please include line numbers in future versions of the manuscript to assist reviewers — it greatly aids clarity and precision in feedback.

We have added line numbers to the Manuscript.

In general referencing in the introduction could do with some more care. For instance, the statement “MAGs constructed from short reads frequently suffer from incompleteness, contamination and high Fragmentation.” is unreferenced, and the first mention of metaMDBG is not referenced, etc.

We have addressed this, adding a relevant reference regarding short-read MAG construction and referencing metaMDBG.

“ntigs into metagenome-assembled genomes (MAGs) [1],” - this [1] should be replaced by reference(s) which better represent the state of the field, rather than a manuscript by the same corresponding author.

We have added references to two recent binners, SemiBin2 and ComeBin.

“In contrast, metaFlye produced only fragmented Genomes” - this section could be written more clearly. It isn’t clear which genomes are being referred to here, for instance. Perhaps it would read better if the E. coli strains and low abundance species were discussed separately, rather than everything at once. It would also be clearer if the true abundance table was given, so that the precise meaning of “low abundance” is immediately available to the reader, rather than asking them to chase this down through the Zymo website (an option which may also be not available in the long term).

We have revised this section for clarity, now discussing E. coli strains and low-abundance species separately. The results are also reformatted in Supplementary Table 2 on a per-species basis, so comparisons across assemblers is easier. We also have added the relative abundance of species in this table.

“Significantly” - word should be replaced as no statistical analysis is present (nor should there be).

It is not clear which use of the word significantly the reviewer objects to. In those contexts where confusion about whether a test was performed might occur we have removed it, e.g:

“On this dataset, nanoMDBG **substantially** outperformed state-of-the-art”

“For all three tested real communities, nanoMDBG reconstructed **appreciably** more“

“By performing the correction step in minimizer-space, our novel method, nanoMDBG, naturally extends metaMDBG to ONT data without impacting overall performance.”

“In contrast, DeChat substantially enhanced assembly outcomes: it improved the recovery of scMAGs by metaMDBG”

We also just deleted the word significantly in some other contexts. Otherwise we believe it is clear and valid english.

Capitalisation of bioinformatic tool names should be made more carefully. For instance the first C of CheckM2 is capitalised (there were several variants in the ms), and the tool name is CheckM2, not CheckM.

This is now corrected for CheckM2, CheckV and geNomad. We believe other capitalisations are correct.

The words “near complete” and “complete” are used to define categories of MAG reconstruction accuracy. However, these words have precise meaning as defined in the MIMAG standard, and they should not be changed to suit a particular study. They also incorporate genome characters beyond CheckM2-based completeness and contamination statistics. For instance, requirements to include rRNA operons. One option might be to use one threshold of contamination (say 5%), and then the categories names can reflect different levels of completeness. This way more subtle trends than high quality vs medium quality can be showcased, while not confusing the precise categorisations set up in the MIMAG standard.

We respectfully disagree with the reviewer here. We are not obliged to use the MIMAG definitions and many papers in the assembly/binning literature do not. In fact there does not appear to be a definition of near-complete in MIMAG but I assume the reviewer is referring to high-quality draft. We would suggest that as long as we are clear in our definitions then it is a fair comparison across assemblers. We do though now use a single contamination category of 5% so that our definitions only reflect completeness as the reviewer suggests. We give the proportions of near-complete MAGs that contain a full complement of RNA genes elsewhere (it is around 90%) which makes it clear that this is not used in our completeness definition.

“Not every read will be represented in the assembly, .. “ - I did not understand the logic for excluding unassembled reads from this percentage. I suggest rephrasing or removing this interpretation.

We have rephrased this paragraph lines 157 - 162 to better explain why comparing reads mapping to MAGs to reads mapping to the assembly is useful. Hopefully this is clearer now.

“We note that we ..” over-use of “we” here. There is an argument also that is true more generally in the manuscript, but perhaps that is a matter of taste.

We removed the two phrases ‘We note that we’ but otherwise we feel that ‘we’ is acceptable usage in a multi-author scientific manuscript.

“Varying sequencing depths”: On first read I worried about non-equal read quantities being used, which might bias the results. Further reading allayed my concerns on this, but perhaps this sentence could be clearer on that point up front.

Well we see the reviewers point but equal read numbers would be less valid in our opinion. Since it would favour longer reads. We have added a clarification about this though.

‘Note that different average read lengths may result in differing read numbers at the same depth when comparing between technologies but this in our opinion is the most valid comparison.’

“The improved MAG recovery by nanoMDBG translates into a more representative picture of microbial diversity at all levels of evolutionary divergence.” While this is likely true, it cannot be concluded so confidently, because the true diversity remains unknown.

We have now clarified that this is ‘compared to the other assemblers tested.’

“Specific” vs “shared” - are these the same meaning? If so only 1 should be used to avoid confusion.

No, "shared" refers to MAGs that are recovered by both HiFi and Nanopore technologies, while "specific" (or unshared) refers to MAGs that are recovered by only one of the two technologies.

“Gut” or “human gut”? Only one descriptor should be used, to avoid confusion.

We have replaced Gut with Human gut.

“Proved essential”. It may not have been essential, even if the manuscript shows it helps assembly quality. Suggest rephrase.

We have rephrased this, removing ‘proved essential’ to give “Our new correction module greatly improved the number of reconstructed MAGs, phages and plasmids compared to metaMDBG.”

“The latest ONT sequencing technology now produces results comparable to PacBio HiFi sequencing at equivalent sequencing depth, despite the remaining differences in raw read accuracy.” - this is true only if the per-base accuracy of the contigs is comparable, which has not been established.

This is a fair point in the new section “Evaluating metagenomics assembly errors” we show that for soil at least that there are still more frame shift errors in ONT contigs than HiFi we have adjusted this statement accordingly. To give:

“Abstract: As a result of these advances, we show that the latest ONT technology can now produce comparable MAG construction results as those obtained using PacBio HiFi sequencing at the same sequencing depth.”

“Line 86: .. we demonstrate that nanoMDBG can now deliver ONT-derived results comparable in MAG number to those achieved with HiFi sequencing.”

“read Metagenomicss” - error

Fixed.

“adapt our algorithm to a streaming assembler, “ - grammatical mistake. This sentence is also too long.

We have reworded this:

“Line 364: In future, it may prove useful to implement a version of our algorithm that can perform streaming assembly, assembling reads ...”

“The most similar reads (up to approximately 20x coverage)”. This should be qualified to mean that they are the most similar in (sparse) minimizer space, which is not the same thing.

We have updated this sentence as follows:

“Line 444: For each target read, we recruit the most similar reads in (sparse) minimizer-space (up to approximately 20x coverage) using seed-and-chaining with a very-low density of minimizers.”

“Filitered” - grammatical mistake.

We could not find this error in the MS.

“And”: should not start a sentence.

Fixed.

“Perform against” - grammar mistake.

Fixed.

“Generating consensus” - this section needs more detail.

We have added a detailed description of the method used to extract the most supported path in this section.

“Line 485: ... After all the alignments have been added to the variation graph, we extract the most supported path from the graph as a consensus using the following procedure. We begin by initializing a hash table S that maps each node v in the graph to its support score, with all values initially set to zero: $S[v] = 0$. Next, we perform a topological sort of the graph’s nodes and iterate through them in the sorted order. For each node v , we identify the highest-supported incoming edge $e = (u, v)$ with weight w_e , and update its support score as $S[v] = S[u] + w_e$. The node u is recorded as the parent of v for path reconstruction. The consensus path is then constructed by identifying the node v_{max} with the highest support score $S[v]$, and backtracking through its parent nodes until a node with no parent is reached.”

“At each multi-k” - is this multi-k, or just each k?

We have revised this sentence for clarity as follows: At each iteration of the multi-k algorithm

A description of the hardware architecture used should be given, if walltime is the main output of the benchmark. How was it compiled? With optimisations for the specific hardware or a general-purpose build for e.g. conda?

We have added information about the system architecture and software in the Methods section as follows:

“Line 555: We ran all assemblers with 32 CPU threads on a machine equipped with a 2.60 GHz Intel(R) Xeon(R) Gold 6132 CPU with 112 cores and 2,038 GB of memory. All programs were installed via Bioconda.”

“-map-hifi” is not an option of minimap2

Fixed: -x map-hifi

“checkV” - which version?

Fixed: checkV (v.1.0.3)

“Assessment of completeness..” Does this section refer only to the Zymo community? If so, it would be clearer to say that directly.

We have renamed this section to “Quality assessment of assemblies on the Zymo mock community”

Emdashes - in several places it appears “--” has been replaced by an emdash, which is incorrect if they refer to command line options.

We have fixed this formatting issues.

Figure 4 “showcase” - Consider replacing ‘showcase’ with a more precise verb like ‘illustrates’ or ‘demonstrates’ to maintain academic tone.

Fixed.

“The highest sequencing depth available” - this was unclear to me - 50, 200 and 250G respectively? Or the full depth available for each method for each assembler?

We used 50 Gb of Human Gut, 200 Gb of Zymo Fecal Reference and 250 Gb of Soil. We have added this information in the legend:

“We de-replicated MAGs per dataset using the highest sequencing depth available for each dataset (50 Gb, 200 Gb and 250 Gb respectively).”

Figure S3 “assemblers” should be singular. Y-axis Virus should be plural.
Gbps - the s is not needed.

Fixed.

We thank the reviewers for reading the revised version of our manuscript. This time there were relatively minor comments to address but we did repeat the IDEEL analysis with a cut-off of 5 kb as recommended by Reviewer 4 and add some additional description of the methods used in the error analysis as requested by them.

We include a point-by-point response to the reviewers below with our responses highlighted in blue. We hope that our manuscript is now suitable for publication.

Yours sincerely,

Dr Christopher Quince

Group Leader
Earlham Institute

REVIEWER COMMENTS

Reviewer #1 (Remarks to the Author):

The authors have addressed the comments from my previous review of this manuscript.

One minor typo in the Introduction: 'The initial step in many metagenomic analysis' should be 'The initial step in many metagenomic analyses'.

Fixed.

Reviewer #2 (Remarks to the Author):

The authors addressed our comments and made the appropriate changes to the text where necessary, including additional discussions, explanations and analyses. We have only a couple minor comments below that do not require re-review.

Proportion of read correction time for the same dataset in lines 82 and 322 (10% and 6%) are not consistent.

Fixed.

We appreciated the expanded discussion on the issue of strain assemblies. The authors mentioned that metaFlye attempts to reconstruct strain genomes, but the more recent work from this group (<https://pubmed.ncbi.nlm.nih.gov/39327484/>) might be more relevant in this context.

We agree, we have added a reference to strainy to this discussion, noting that it maps onto assembly graphs rather than genomes (see Line 359).

Reviewer #3 (Remarks to the Author):

Reviewer #4 (Remarks to the Author):

In responding to these reviews, the authors have made substantial improvements that greatly benefit the manuscript.

In the IDEEL comparison, I am curious what the effect of a more fragmented assembly might have on the results. In particular, if many contigs are short, then ORFs on the ends of contigs might be substantial in number. I note that the -c parameter of prodigal is not used in IDEEL (maybe it should be, though of course that is not the tool being assessed here). The authors might choose to restrict the IDEEL analysis to contigs over a certain length, or even compare between different assemblers just those contigs in length bins e.g. 5-10kb, 2-5kb, etc. It is particularly a problem if one assembler uses a different minimum output contig length threshold - I have not looked into this.

We have repeated the IDEEL analysis using a 5 kb contig length threshold and have updated the text, supplementary materials and Figure 5 accordingly. Applying this length filter had only a minimal impact on the results. This is largely because the existing 10x coverage filter had already excluded the majority of short contigs in each assembly.

The methods for the next section appear to be incomplete. For instance, it was not explained how clipping counts were determined.

We have added a few more details on this in the main text (Lines 290-291) and a whole new section to the methods (Lines 612-620) 'Identification of errors in assemblies' which details how this and the other error evaluations were performed.

273: It was not clear from the text what datasets were used for the first long contigs benchmark. I suppose it is the union of all contigs from all datasets assembled separately?

Yes it was the union of all three assemblies Human Gut, Zymo Fecal Reference and Soil. We have clarified that in the caption to Figure 5.

228-9: Should be past tense.

We could not find the phrase the reviewer is referring to here. This section appears to all be in the past tense already.